# Proteomic-Based Analysis of Hypoxia- and Physioxia-Responsive Proteins and Pathways in Diffuse Large B-Cell Lymphoma

**DOI:** 10.3390/cells10082025

**Published:** 2021-08-08

**Authors:** Kamila Duś-Szachniewicz, Katarzyna Gdesz-Birula, Krzysztof Zduniak, Jacek R. Wiśniewski

**Affiliations:** 1Department of Clinical and Experimental Pathology, Institute of General and Experimental Pathology, Wrocław Medical University, Marcinkowskiego 1, 50-368 Wrocław, Poland; katarzyna.gdesz-birula@student.umed.wroc.pl (K.G.-B.); krzysztof.zduniak@umed.wroc.pl (K.Z.); 2Biochemical Proteomics Group, Department of Proteomics and Signal Transduction, Max Planck Institute of Biochemistry, 82152 Martinsried, Germany; jwisniew@biochem.mpg.de

**Keywords:** hypoxia, physioxia, diffuse large B-cell lymphoma (DLBCL), B-cell non-Hodgkin lymphomas (B-NHL), label-free quantitative proteomics, protein–protein interaction network (PPIN), cell stress

## Abstract

Hypoxia is a common feature in most tumors, including hematological malignancies. There is a lack of studies on hypoxia- and physioxia-induced global proteome changes in lymphoma. Here, we sought to explore how the proteome of diffuse large B-cell lymphoma (DLBCL) changes when cells are exposed to acute hypoxic stress (1% of O_2_) and physioxia (5% of O_2_) for a long-time. A total of 8239 proteins were identified by LC–MS/MS, of which 718, 513, and 486 had significant changes, in abundance, in the Ri-1, U2904, and U2932 cell lines, respectively. We observed that changes in B-NHL proteome profiles induced by hypoxia and physioxia were quantitatively similar in each cell line; however, differentially abundant proteins (DAPs) were specific to a certain cell line. A significant downregulation of several ribosome proteins indicated a translational inhibition of new ribosome protein synthesis in hypoxia, what was confirmed in a pathway enrichment analysis. In addition, downregulated proteins highlighted the altered cell cycle, metabolism, and interferon signaling. As expected, the enrichment of upregulated proteins revealed terms related to metabolism, HIF1 signaling, and response to oxidative stress. In accordance to our results, physioxia induced weaker changes in the protein abundance when compared to those induced by hypoxia. Our data provide new evidence for understanding mechanisms by which DLBCL cells respond to a variable oxygen level. Furthermore, this study reveals multiple hypoxia-responsive proteins showing an altered abundance in hypoxic and physioxic DLBCL. It remains to be investigated whether changes in the proteomes of DLBCL under normoxia and physioxia have functional consequences on lymphoma development and progression.

## 1. Introduction

The environment in which lymphoma cells are found in the human body has different levels of oxygenation. Lymphatic tissues show a lower concentration of oxygen than blood (0.6–2.8% of oxygen) [1,2], while the oxygen concentration in bone marrow counts for approximately 1.3–4.2% [3]. Indeed, secondary lymphoid organs, such as the spleen, may also encounter a hypoxic environment (0.5–4.5% of oxygen) [4]. When normal B-cells are incapable of adjusting to low O_2_ and do not survive hypoxic stress in vitro for more than 48 h, lymphoma cells respond to hypoxic stimuli through alterations in gene expression and escape from cell death [5]. Surprisingly, the exact molecular background of the adaptation to an imbalance in oxygen supply is still not fully understood.

Mammalian cells have historically grown and divided well in ambient air, thus culturing cells in a more physiologic setting concerning oxygen (so-called physioxia) has not received approval for many years. Meanwhile, with technological advances, data support that cells benefit from being cultured in physioxia. There is strong evidence that physioxia influences the growth [6,7,8], differentiation [9,10], and survival [11] of many cell types, cultured as monolayers and three-dimensional spheroids. Moreover, culturing of cells under physiological oxygen has significant consequences on, e.g., adhesion and motility [12], self-renewal [13], and production of EMC and growth factors [14]. Notably, Timpano et al. reported that physioxic cultures significantly improve mitochondrial metabolic activity while reducing DNA damage [15]. Surprisingly, the data describing the global differences in the transcriptomes and proteomes of cells when cultured in physiological oxygen are lacking. However, the observed differences in the phenotypes between normoxic and physioxic cells raise essential questions on the relevance of the in vitro studies performed in normoxic conditions, primarily when they are related to the development of potentially life-saving therapies.

Non-Hodgkin’s B-cell lymphomas (B-NHLs) are a heterogeneous group of lymphoproliferative malignancies originating from the different stages of B-lymphocyte development and maturation [16]. It was believed that lymphoma cells proliferate independently from the tissue blood supply, and this is partially true for indolent lymphomas, in which limited metabolic requirements are met directly from the circulation [2]. Meanwhile, the overall survival of aggressive lymphomas, including the most frequent diffuse large B-cell lymphoma (DLBCL), strongly depends on continuous tissue oxygenation and nutrient supply [17], similarly to solid tumors. Aggressive lymphomas share other features with solid tumors, including rapid cell growth, angiogenesis, and the presence of focal necrosis; thus, their response to hypoxia could resemble that of solid tumors [18]. However, an increasing number of reports are showing that hypoxia-related changes in DLBCL are more complex and heterogeneous than originally thought [5,19].

The majority of cancers undergo extensive cellular reprogramming to adapt to environmental challenges, such as hypoxia. Low oxygen pressure in the cancer microenvironment leads to the transcriptional induction of several genes, of which key regulators are the hypoxia-inducible transcriptional factors (HIFs) [20]. The HIFs regulate several target genes in diverse biological pathways; thus, cancer cells respond to the hypoxic stress by controlling of mRNA-specific translation [21] and cell cycle, increasing angiogenesis [22], and metabolic reprogramming [23,24]. Recently, Bhalla et al. observed a 50% decrease in translation in DLBCL under the influence of activated HIF-1α [5]. Importantly, translation was not completely stopped, and targets, such as GLUT1, HK2, and CYT-C, proved resistant to translational repression due to hypoxia. In turn, Sharma et al. revealed the hypoxia and normoxia-induced RNA editing events in lymphoma by RNA sequencing, which was enriched for genes involved in mRNA translation and ribosome function [24]. Importantly, the authors showed that inhibition of mitochondria respiration results in Warburg-like metabolic remodeling that occurs independently of HIF-1α. Additional recent observations demonstrate further crosstalk of hypoxia and mitochondrial oxidative phosphorylation [25], mTOR [23,26], and BCR [27] pathways in normal and malignant B-cells.

Few previous works have been dedicated to describing the protein profile of B-NHLs, especially DLBCL [28,29,30]; however, none of these results showed how lymphoma cells would behave under variable oxygen mimicking in vivo settings. Therefore, our goal here was to describe the quantitative changes in global protein expression of DLBCL cells under hypoxic (1% O_2_) and physioxic (5% O_2_) conditions with large-scale LC–MS/MS, and subsequent bioinformatics data analysis. We identified for the first time hundreds of hypoxia- and physioxia-responsive proteins, many of which have not been previously associated with hypoxia. Clearly, the observed response to hypoxia in DLBCL is highly complex and heterogenous, which was previously highlighted in work based on whole-genome sequencing data [19]. At the same time, the enrichment analysis revealed several hypoxia- and physioxia-responsive pathways, commonly altered in lymphoma cell lines, including ribosome biogenesis, translation, metabolism, and cell cycle. Hypoxia largely impacts lymphoma progression by promoting a more malignant phenotype, associated with poor clinical prognosis in several cancer types [31]. Thus, the results presented in this work provide a valuable source of new molecular targets and pathways potentially important for the design of new anti-B-cell therapies of NHL.

## 2. Materials and Methods

### 2.1. Cell Culturing under Physioxia and Hypoxia 

Ri-1, U2904, and U2932 cell lines were obtained from the Leibniz Institute German Collection of Microorganisms and Cell Cultures (DSMZ, Braunschweig, Germany). The characteristic of the cell lines are presented in Table 1.

For the experiments, 1 × 10^6^ of cells were seeded on 75 cm^2^ flask and incubated for 72 h in Gibco™ RPMI 1640 medium with GlutaMAX (Thermo Fisher Scientific, Berlin, Germany) containing 10% heat-inactivated fetal bovine serum (FBS, Thermo Fisher Scientific) in a humidified atmosphere in standard conditions. The viability of diffuse large B-cell lymphoma (DLBCL) cells was evaluated by Trypan blue dye exclusion assay on the Countess Automated Cell Counter (Thermo Fisher Scientific). Cells were routinely >95% viable.

Next, 3 × 10^6^ of cells/mL were subjected to either physioxic (5% O_2_, 5% CO_2_) or hypoxic (1% O_2_, 5% CO_2_) treatment for 72 h using the multigas incubator (New Brunswick Galaxy 48R, Eppendorf, Hamburg, Germany). During incubation, the temperature of 37 °C and proper humidity were maintained. As a control, cell lines incubated in normoxia (21% of O_2_, 5% CO_2_) were used. Four independent biological replicates were conducted for each condition for each cell line. All supplies and media were acclimated for at least 18 h in either hypoxia or physioxia, as previously suggested [32].

### 2.2. Sample Preparation for Mass Spectrometry

For the sample preparation, 1 × 10^6^ of cells were lysed in 150 µL of lysis buffer containing 100 mM of Tris-HCl, pH 8, 50 mM of dithiothreitol (DTT), and 2% of sodium dodecyl sulfate (SDS) for 5 min at 99 °C. Next, lysates were sonicated on the water at room temperature for 2 min and centrifuged at 10,000 RCF for 10 min, followed by transferring supernatant to the new tubes. The total protein concentration was measured in accordance with the tryptophan fluorescence (WF)-based assay [33]. Sample aliquots containing 70 µg of total protein were processed using the multi-enzyme digestion filter aided sample preparation (MED FASP) protocol [34] with minor modifications [35]. Briefly, proteins were consecutively digested overnight with endoproteinase Lys-C (Wako Chemicals, Neuss, Germany) and then with trypsin (Promega, Madison, WI, USA) for 3 h. The enzyme to protein ratio was 1:50. Aliquots containing 10 µg of total peptides were desalted on C18-StageTips [36] and concentrated to a volume of ~5µL. Samples were stored frozen at −20 °C until analysis.

### 2.3. LC–MS/MS

Analysis of peptides was performed using a Q Exactive HF Mass Spectrometer (Thermo-Fisher Scientific). Moreover, 1 µg of total peptides was chromatographed on a 50 cm column with a 75 µm inner diameter packed C18 material (Dr. Maisch GmbH, Ammerbuch, Germany). Next, peptide separation was carried out at 300 nL/min for 90 min with the use of two-step acetonitrile (ACN) gradient of 5–60% over the first 75 min and 60–95% for the following 15 min. The temperature of the column oven was 60 °C. The mass spectrometer operated in data-dependent mode. Survey scans were acquired at a resolution of 50,000 at m/z 400 with transient time 256 ms. Most abundant isotope patterns with charge ≥+2 from the survey scan (300–1650 m/z) were selected up to the top 15, with an isolation window of 1.6 m/z and fragmented by HCD with normalized collision energies of 25. The maximum ion injection times for the survey scan and the MS/MS scans were 20 and 60 ms, respectively. The ion target value for MS1 and MS2 scan modes was set to 3 × 10^6^ and 10^5^, respectively, while the dynamic exclusion was 25 s and 10 ppm. The mass spectrometry data were deposited to the ProteomeXchange Consortium via the PRIDE partner repository [37] with the PXD026726. Reviewer account details: username: reviewer_pxd026726@ebi.ac.uk; password: 6rGpX4.

### 2.4. MS/MS Data Analysis and Statistical Analysis

MaxQuant (MQ) software (Max Planck Institute, Martinsried, Germany) was used for spectra searching. A maximum of two missed cleavages was allowed. Carbamidomethylation of cysteines was set as a fixed modification. The minimum peptide length was specified to be seven amino acids. The initial maximal mass tolerance in MS mode was set to 7 ppm, while fragment mass tolerance was set to 20 ppm for HCD data. The maximum false peptide and protein discovery rate was set as 0.01. Specific protein concentrations were calculated by the “Total Protein Approach” (TPA) [38] using raw intensity MQ output. Statistical analysis of proteomic data was conducted by Perseus software v.1.6.10.45 (Max Planck Institute for Biochemistry) [39]. Importantly, the samples were filtered for proteins detected in 100% in each group. The absolute protein concentration was log2-transformed, z-scored and the two-sample T-test was performed. Benjamini–Hochberg FDR procedure was used to calculate *p*-values, and *p* < 0.05 was considered statistically significant. The following comparisons were performed: hypoxic cell lines versus corresponding normoxic cell lines, physioxic cell lines versus corresponding normoxic cell lines, and hypoxic cell lines versus corresponding physioxic cell lines to identify the hypoxia- and physioxia-related proteins. The ratio was calculated by (1) dividing the mean TPA values of the hypoxic/physioxic cell lines with the mean TPA values of the corresponding cell lines under normoxia; and (2) dividing the mean TPA values of the hypoxic cell lines with the mean TPA values of the corresponding cell lines under physioxia. The cutoff values of ±1.5-fold for up- and downregulated proteins between samples were established.

### 2.5. Bioinformatic Analysis

The differentially abundant proteins were analyzed by open-source bioinformatics software platform Cytoscape (version 3.8.2). Functional interaction network analysis was performed using ClueGO and CluePedia (version 2.5.7), STRING, and cytoHubba Cytoscape plugins [40,41,42]. Pathway enrichment analysis was carried out and visualized with the Metascape [43]. Protein–protein interaction network for up- and downregulated proteins was visualized by ClueGO against Gene Ontology (GO)–Biological Processes (all updated on 8 May 2020) with standard settings. Only pathways with *p* < 0.01 were shown. The Venn diagrams were created with the web tool from Bioinformatics and Evolutionary Genomics group [44].

## 3. Results

### 3.1. Overview of Data

We were interested in determining how hypoxia (1% of oxygen) and physioxia (5% of oxygen) differentially impacts the proteomes of DLBCL. We first cultured Ri-1, U2904, and U2932 cells under the hypoxic and the physioxic condition for 72 h. Cells maintained under the normoxic conditions served as a control. The experiment was performed in four biological replicates for each cell line in each condition. The findings of our comparative LC–MS/MS proteomic study are summarized in Figure 1. A total of 8239 proteins were identified across all three cell lines, of which 7959, 8031, and 8020 were found in Ri-1, U2932, and U2904 cells, respectively (Figure 1A).

Importantly, principal component analysis (PCA) differentiated the cell lines treated under normoxia, physioxia, and hypoxia, and revealed similarity between Ri-1 and U2904 samples, especially under normoxia and physioxia (Figure 1B). In turn, Figure 1C shows an unsupervised hierarchical clustering dendrogram of all U2932 samples. Two main clusters are found, first representing cells under hypoxia, and second grouping samples under physioxia and normoxia. Next, we analyzed the Pearson’s correlation coefficient for each cell line and the representative results for a U2932 analysis are presented in Appendix A. Biological replicates performed for each experimental condition showed high Pearson’s correlation coefficient values, with a maximum of 0.92. For protein identification data, see Appendix A.

### 3.2. DLBCL Shows Extensive Changes in Proteomic Profile under Hypoxia and Physioxia When Compared to Normoxia

We analyzed differentially expressed proteins with ≥1.5-fold (higher or lower) differences in the protein abundance and *p*-values *p* < 0.05 under hypoxia and physioxia in comparisons to 21% of oxygen. Compared with the cells cultured under normoxic conditions, under hypoxia we identified 718, 513, and 486 of differentially abundant proteins (DAPs) in Ri-1, U2904, and U2932 cell lines, respectively. Thus, Ri-1 cell line appeared to be the most hypoxia-responsive with 11.45% of proteins with an altered abundance. U2904 and U2932 cells exhibited a relatively lower response to this treatment—the expression of 8.14% and 7.6% of proteins was significantly affected. Maximum alterations of over 195-fold decrease in protein abundance were found for AIM1 and CDK16 in hypoxic Ri-1 cells. In turn, the most upregulated protein in our dataset was RAB39B in hypoxic U2905 cells with a fold change of 30.06. For more details, see the complete lists of DAPs identified in each cell line (Appendix A). Next, we used a volcano plot to visualize DAPs identified in hypoxia (Figure 2A,D,G) and physioxia (Figure 2B,E,H).

Our data showed that the number of upregulated proteins was similar between the three cell lines, while the number of downregulated proteins was significantly higher in Ri-1 cells. Each cell line under physioxia exhibited weaker changes in the protein abundance when compared to those induced by hypoxia, and the percentage of DAPs was 5.3%, 4.37%, and 4.29% for U2904, Ri-1, and U2932, respectively. A full list of hypoxia, and physioxia-responsive proteins identified in each particular cell line is available as Appendix A. Furthermore, Table 2 shows the confirmed HIF-1α targets identified in this study, of which ALDH6A1 was upregulated in each cell line under hypoxia.

The numbers of DAPs overlapping in hypoxia and physioxia for each cell line separately are presented in the Venn diagram (Figure 2C,F,I). The 49.29%, 40.58%, and 37.2% of DAPs identified in physioxia were common with those identified in hypoxia in Ri-1, U2904, and U2932 cells, respectively. In turn, the 19.22%, 21.83%, and 26.33% of altered proteins in hypoxia were shared with DAPs identified in physioxia. This suggests that distinct mechanisms may be responsible for the cellular response to hypoxia and physioxia. Next, we classified DAPs in relation to fold change ranges, Figure 3A. The majority of DAPs were up- or downregulated at a fold change range between 1.5 and 2.5. The percentage of the most upregulated proteins with fold change > 5.0 identified in each study group varies from 8% to 15.94% for physioxic U2932 cells and hypoxic Ri-1 cells, respectively. On the other hand, the percentage of the most downregulated proteins (fold change >−5.0) is the highest for physioxic U2932 cells (14.3%) and the lowest for hypoxic U2904 cells (5.7%). The lists of overlapping proteins in each cell line are provided in Appendix A.

#### Response to Both: Hypoxia and Physioxia Is Cell Line Specific

Then, we compared DAPs identified in each cell line, after hypoxic and physioxic treatment, and the analysis revealed that the response to increased oxygen is cell line specific. Among the differentially abundant proteins in hypoxia, 518, 382, and 315 proteins were specifically expressed in cell Ri-1, U2932, and U2904 cell lines, respectively (Figure 3B). In turn, the 110 of DAPs were shared in hypoxia between Ri-1 and U2904 cells, while 70 and 40 DAPs were common for Ri-1 and U2932, and U2904 and U2932, respectively.

Only 19 of DAPs were common in the three compared groups, including TRUB2, SEC14L1, CLPTM1, PICALM, ALDH6A1, CCDC86, PPT1, AK4, MMAA, VIMP, HMGCL, GTPBP10, SGSH, EIF4A2, CFDP1, METTL17, KDM3A, DDX56, and PPP1R7. Under physioxia, 280, 227, 213 of DAPs were altered specifically in U2904, U2932, and Ri-1 cells respectively (Figure 3C). 40 of DAPs were overlapping in Ri-1 and U2904 cells, while 21 and 25 of the DAPs were shared in physioxia between Ri-1 and U2932, and U2932 and U2904 cells, respectively. Two altered proteins, CARS2 and METTL17, were identified in all cell lines after physioxic treatment. These findings suggested that the response to physioxia is highly cell line specific. The complete list of shared proteins is presented as Appendix A.

### 3.3. Hypoxic Treatment Exhibit Lower Magnitude of Changes When Compared with Physioxia

Next, we analyzed the cell lines proteomes under hypoxia versus corresponding cells cultured under physioxia. We identified the 320, 279, and 357 DAPs in Ri-1, U2904, and U2932 cells, respectively (Figure 4A–C). Importantly, we observed that 47.81%, 31.18%, and 30.81% of DAPs identified in hypoxic Ri-1, U2904, and U2932 cells, respectively, were shared with DAPs identified in corresponding cell lines when hypoxia compared to normoxia (Figure 4D).

As expected, the number of DAPs were significantly lower than the number of DAPs identified in the hypoxia compared to normoxia. The differences in the number of outliers identified in hypoxic cell lines ranged 55.43%, 42.59%, and 30.41% of the decrease in the number of DAPs in Ri-1, U2904, and U2932 cells, respectively. Regarding the fold change values of DAPs, these do not exceed a 10-fold increase in protein abundance in any cell line. CRYZ was the most upregulated protein in our dataset, with a fold change of 9.545 in U2904 cells. In turn, the highest fold decrease of 48.331 was noticed for LRRC15 in U2932 cells.

Response to hypoxia, when compare with physioxia, is cell line-specific (Figure E). 308, 257, and 221 of outliers were specifically expressed in U2932, Ri-1, and U2904 cell lines, respectively. In turn, only 35 DAPs were shared in hypoxia between Ri-1 and U2904 cells, while 26 and 21 DAPs were common for Ri-1 and U2932, and U2904 and U2932, respectively. Two altered proteins, KDM3A and MYO16, were identified in all cell lines under hypoxia. A complete list of hypoxia-responsive proteins identified in each cell line compared to physioxia is available as Appendix A.

### 3.4. Functional Enrichment Analysis

#### 3.4.1. Upregulated Proteins Reveals Changes in Metabolism and HIF-1 Signaling Pathway under Hypoxia

To find out which biological pathways are regulated preferentially in DLBCL cells by hypoxia and physioxia relative to normoxia, an enrichment analysis was performed with the use of Metascape. The enrichment was done for up- and downregulated proteins separately for each cell line (12 enrichment analyses in total). The top 20 clusters with their representative enriched terms (one per cluster) were presented for each experimental condition (Figure 5 and Figure 6). As expected, over-represented proteins in DLBCL cells under the hypoxic conditions were associated mostly with amino acid and nucleotide metabolic processes, and HIF1 signaling pathway (Figure 5A,C,E). Proteins involved in the response to oxidative stress have also a higher abundance in hypoxia and physioxia relative to normoxia. Our data, furthermore, showed that in physioxia, most of pathways were predominantly affected within a particular cell line, e.g., SUMOylation of transcription factors in Ri-1 cells, TNF alpha signaling in U2904 cells, or chaperone mediated protein folding in U2932 cells (Figure 5B,D,F).

#### 3.4.2. Ribosome Biogenesis, Translation, and Mitochondrial Gene Expression Decrease in in DLBCL under Hypoxia and Physioxia

In turn, pathway mapping shows a decrease in abundance of proteins mainly involved in the ribosome biogenesis, translation, and mitochondrial gene expression (Figure 6A–F). Additionally, pathways associated with the cell cycle, e.g., negative regulation of cell cycle, chromosome segregation or G2/M transition were enriched among downregulated proteins, particularly in U2932 cell line. Interestingly, hypoxia also led to downregulation of interferon signaling in Ri-1 and U2904 cells, which is important for the success of anticancer treatments. According to our data, long-term physioxia downregulated similar pathways when compared to those perturbed by hypoxia, however, the observed changes were lower, what may results from a significantly lower number of physioxia-responsive proteins (Figure 6B,D,F). Physioxic treatment significantly dysregulated ribosome biogenesis in Ri-1 and U2904 cells. In turn, mitochondrial metabolism processes were affected in Ri-1 and U2932 cells. Interestingly, regulation of cell-adhesion and integrin-mediated signaling pathways were affected in U2904 cells, what we previously described in the optical-tweezers-based study (Figure 6D).

### 3.5. Hypoxia-Responsive Proteins Make a Complex Protein–Protein Interaction Network in DLBCL

Ri-1 cell line with 718 DAPs was the most affected by hypoxia; thus, we decided to study the protein–protein interaction network on this representative. The up- and downregulated proteins were analyzed separately by the use of ClueGO application of Cytoscape (v.3.8.2) in accordance with Gene Ontology–Biological Processes to visualize the protein–protein interaction network (Figure 7A,B). For the list of proteins belonging to all enriched pathways, see Appendix A.

Six function cluster groups related to the response to hypoxia, amino acid metabolic process (particularly valine), histone demethylase activity, deoxyribonucleotide metabolic process, microtubule polymerization, as well as the oxidative phosphorylation were enriched from the upregulated proteins. Only two clusters related to the cellular response to hypoxia and amino acid metabolic process were functionally connected to each other (Appendix A) and the shared upregulated proteins were as followed: PSMA3, PSMB5, PSMC1, PSMC3, and PSMD11, PSMD13. Additionally, the most DAPs have been assigned to the above pathways, namely 19 and 11 of proteins belonging to the amino acid metabolic processes and the cellular response to hypoxia, respectively. In general, seven of the clusters were enriched by equal or more than 10 proteins. Importantly, our data showed upregulation of several signaling pathways with big importance to lymphoma pathogenesis, including Wnt-signaling pathway, non-canonical Wnt-signaling pathway, NIK/NF-kappa B signaling, and interleukin 1-mediated signaling pathway.

In turn, the PPIN between downregulated proteins was more complex, and five out of eight main clusters were functionally related. As expected, the proteins with a decreased abundance under hypoxia formed the following clusters: ribosome biogenesis (including 18 pathways), mitochondrial gene expression (13 pathways), cellular metabolic processing (16 pathways), and RNA processing (nine pathways). The other biological processes enriched by ClueGO were those associated with RNA modification, mitotic cell cycle, regulation of transcription, and nucleus organization. The eight pathways were represented by more than 200 proteins, including the cellular nitrogen compound metabolic process with 276 of proteins assigned. Moreover, 51, 92, 46, and 21 of the downregulated proteins were assigned to the pathway of ribosome biogenesis, RNA processing, translation, and mitochondrial gene expression, respectively. In the PPIN of downregulated DAPs, several proteins overlapped between the distinct pathways, as visualized in Appendix A using the CluePedia app from Cytoscape.

### 3.6. Hypoxia Induces Prominent Changes in Translation in DLBCL

Since we were interested in pathways predominantly affected by hypoxia, we analyzed the translation pathway (R-HSA-72766) in hypoxic Ri-1 cells in more detail. DAPs belonging to the above pathway in accordance with the Metascape enrichment analysis were uploaded to STRING to create the PPIN, Figure 8A. The changes to translation-related proteins were suggestive of both promotion and inhibition of this process. Among the 14 and 20 up- and downregulated proteins, respectively, eukaryotic translation initiation factors (EIFs) are well known as hypoxia-related proteins. Some of the DAPs related to translation were weakly described in hypoxia, including mitochondrial large subunit proteins (MRPs), VARS, and DDX6, while e.g., PARS2, MARS2, and ERAL1 proteins have not been associated with hypoxia so far. In particular, we have demonstrated the impairments in the expression of MRPs under hypoxia. Several of mitochondrial large subunit proteins (MRPL11, MRPL14, MRPL21, MRPL14, MRPL28, MRPL37, MRPL42, MRPL53) were found to be downregulated, while three mitochondrial small subunit proteins (MRPS11, MRPS12, MRPS14) were significantly upregulated. Importantly, both clusters of proteins (involved in the ribosomal translation) closely and functionally interact with each other. Furthermore, our data show an altered abundance in functionally connected ribosomal proteins, including RPS5, RPS3A, RPS29, RPS6KB1, RPL6, and RPL32.

### 3.7. Hub Proteins Associated with Hypoxia and Physioxia Are Frequently Related to Ribosome Biogenesis

Finally, we used the cytoHubba plugin of Cytoscape to rank proteins in a network and find the hub proteins affected the most by hypoxia and physioxia. First, a STRING analysis was performed for DAPs identified in each cell line, and protein–protein interaction relationship network tables were downloaded and visualized using Cytoscape. Next, the hub proteins were identified in accordance with the degree algorithm of cytoHubba. The selected “hypoxic” hub proteins differed between cell lines, but some of them were overlapping including DDX56, MRTO4, RPF2, RSL24D1, and SDAD1 for Ri-1 and U2904 cells, or PWP1, shared between U2904 and U2932 cells. Interestingly, the vast majority of hub proteins are those downregulated in hypoxia. The PPIN between the top 20 hypoxia-responsive proteins are presented for each cell line in Figure 8B,C,D.

Furthermore, most of the Ri-1 hub proteins are related to ribosome biogenesis (GLTSCR2, BRIX1, BYSL), and rRNA processing (e.g., RRP9, IMP3, and NSA2). CCNB1 and PLK1 were annotated to the cell cycle. U2904 hub proteins are involved in ribosome biogenesis (e.g., BMS1, PWP1, and TSR1), rRNA processing (e.g., EMG1, NOP14) and nucleotide binding (e.g., DDX21, DDX28, DDX37, and DDX56). In turn, 13 of the U2932 hub proteins were involved in the cell cycle (mainly mitosis), e.g., CDCA5, CDC23, CDC45). Importantly, CHEK1 and RRM2 hubs were annotated to the p53 signaling pathway. For hypoxic Ri-1 and U2904 cells, most hub proteins are those related to ribosome biogenesis and rRNA processing. In turn, hub proteins selected among all DAPs identified in U2932 are mostly related to the cell cycle.

Beyond the hypoxia-related hub proteins listed above, some targets identified in physioxia are worth highlighting. The Appendix A shows hub proteins specifically selected for each cell line after physioxic treatment. The majority of hub targets were related to ribosome biogenesis and translation (Ri-1), cell cycle (U2904), oxidative phosphorylation (U2932), and apoptosis (U2904 and U2932). Our data furthermore showed that the majority of hubs identified in U2932 cells under physioxia are NADH dehydrogenase (ubiquinone) proteins (NDUFs), including NDUFA7, NDUFA8, NDUFA9, NDUFA10, NDUFB1, NDUFB4, NDUFS1, NDUFS4, and NDUFS6 subunits (Appendix A). Interestingly, all of the identified NDUFs have increased in abundance in physioxia. Next, we searched for the overlapping hub proteins identified in certain cell lines under hypoxia and physioxia. GLTSCR2 and RPS3 were common for Ri-1 cells, while UBE2C and RFC4 were found overlapping in U2932 cells. No commonly identified hubs were found for U2904 cells.

## 4. Discussion

Proteomic studies of hypoxia have been carried out extensively in recent years; however, the global effect of hypoxia-driven changes on lymphoma protein profiles remains largely undocumented. In this work, we present a report on global protein profiling in DLBCL cell lines exposed to long-term hypoxia (1%) and physioxia (5%). Since there are a lack of data explaining the role of hypoxia in lymphoma, we aimed to describe the complex molecular response of lymphoma cells to low oxygen concentrations. To the best of our knowledge, this study was the first to assess and compare DLBCL proteome changes under hypoxia, and physioxia. Recently, we better understood the need to study the realistic pattern of temporally variable oxygen exposure to the cells and tissues. In our work, the cells were incubated under constant hypoxia or physioxia for 72 h, while in cancer, the fluctuation in oxygen concentration occurred at irregular intervals with sporadic reoxygenation periods because of dysfunctional tumor vascularity and heterogenic blood supply [17]. In addition, hypoxia does not affect all cells within the tumor evenly. While ideally controlled laboratory experiments should mimic physiological conditions, the knowledge about the real pattern of oxygen gradient within lymphoma and other hematological malignancies is limited. Thus, we presented the first, robust analysis to address the differential lymphoma response to physiological and hypoxic oxygen exposure. Even though the experimental conditions are not ideal, this study was planned and performed following currently applicable standards.

Large-scale LC–MS/MS proteomics was applied to comprehensively characterize the hypoxia- and physioxia-responsive proteins of three DLBCL cell lines: Ri-1, U2932, and U2904, when compared to normoxia. Additionally, we showed the differences in the obtained results when hypoxia was analyzed versus physioxia. In accordance with our results, the long-term increase of oxygen concentration affected the protein profiles in each cell line; however, the response to this treatment was cell line specific. Ri-1 and U2904 cell lines appeared to be the most hypoxia- and physioxia-responsive, respectively. Hypoxia significantly changed the abundance of 718 proteins in Ri-1 cells, while physioxia affected the expression of 280 proteins in U2932 cells compared to normoxia. The vast majority of DAPs were identified within the particular cells and only 19 and 2 of the DAPs were shared between cell lines exposed to hypoxia and physioxia, respectively. Most of these proteins have been recently recognized for their role in carcinogenesis and targeted cancer therapy, including, e.g., EIF4A2 [45], ALDH6A1 [46], SEC14L1 [47], METTL17 [48], KDM3A [49], PPT1 [50], SGSH [51], AK4 [52], and PICALM [53]. Importantly, some of the notable DAPs were recognized as hypoxia-responsive in human cancers, of which EIF4A2 is the most documented [54]. Significantly increased AK4 abundance was detected in lung cancer cells exposed to hypoxia, which is relevant to our work [52]. Moreover, it was noticed that AK4 exaggerates HIF-1α protein expression under hypoxia, leading to endothelial to mesenchymal transition in lung cancer. In turn, lysosomal protein PPT1 was upregulated in colon cancer after prolonged hypoxia [55], while ALDH6A was upregulated in, e.g., breast cancer under hypoxia [32]. In our dataset, the METTL17 was the one DAP overlapping under the hypoxic and physioxic treatment.

Usually, in hypoxia and physioxia driven studies, cells are cultured at specific oxygen concentrations relative to normoxia. Since hypoxia is much closer to the physiologic range than normoxia, we analyzed the changes in the hypoxic proteomes of DLBCL cell lines when analyzed in comparison to physioxia. As expected, the lower quantitative changes in the proteomes of DLBCL cell lines were observed with a maximum decrease of 55% in the number of DAPs identified in Ri-1 cells. Notably, 47.81%, 31.18%, and 30.81% of DAPs identified in hypoxic Ri-1, U2904, and U2932 cells, respectively, were shared with DAPs identified in corresponding cell lines, when hypoxia compared to normoxia. Similar to previously performed comparisons, we observed that the response to hypoxic treatment is cell line-specific, and the majority of DAPs were found within the particular cells. Only two DAPs were shared in tree cell lines under hypoxia vs. physioxia: KDM3A, and MYO16, of which KDM3A was found to be the target of HIF-1α [56].

Heterogeneity of cancers within response to hypoxia, which we reported here, was frequently underlined. Recently, Bhandari et al., based on whole genome sequencing data, quantified hypoxia in 1188 tumors spanning 27 cancer types [19]. The authors observed that inter-tumoral variability in hypoxia was especially elevated in some tumor types, including mature B-cell lymphomas. This contrasted with another hematological malignancy, chronic lymphocytic leukemia (CLL), where little heterogeneity in hypoxia was observed.

The enrichment analysis was performed to annotate the pathways and biological function to DAPs, and subsequently to determine which of them are impaired under hypoxia and physioxia. Knowledge about the perturbed pathways and function can help to understand how DLBCL cells adapt to oxygen stress, and to potentially develop new treatment options. Translation and translational regulation are also pivotal for inducing adaptive stress responses of cancer cells to environmental hypoxia by regulation of gene expression [57]. Cancer cells, including lymphoma, under hypoxia, exhibit global shutdown or reprogramming of translation to promote recovery from stress or cell death [58]. Translation is one pathway that is crucial to cancer development and progression, and surprisingly, the impact of hypoxia on global translation in B-NHL remains largely undocumented. Bhalla et al. observed an oxygen-regulated switch in the protein synthesis machinery [5], as was partially confirmed in our work. The authors measured the protein translation efficiency in several DLBCL cell lines using 35S-labeled methionine incorporation, and revealed a ≥50% reduction in translation upon activation of HIF1α. Furthermore, it was observed that normal primary B-cells did not survive prolonged hypoxic stress when exposed to 1% hypoxia for 24 to 48 h. Recently, Sharma et al. demonstrated changes in translational and ribosomal genes in B-NHL under hypoxia [24], including genes encoding eukaryotic initiation factor complexes (EIFs), which is in line with our work. Under physiological conditions, EIFs are involved in all molecular aspects of translation initiation in mammals. Importantly, extensive research in the past two decades has indicated that EIFs are implicated in various types of cancer [59,60], including DLBCL [61]. Consistently with these reports, we identified several EIFs that are dysregulated in DLBCL under hypoxia. EIF2S1, EIF3F, and EIF4A2 were found upregulated in Ri-1 cells, while EIF3K and EIF4G1 were downregulated. Importantly, EIF4A2 was overlapping in all three cell lines in our dataset. Recently, eIF4A2 was indicated to be a regulator of hypoxic translation and colorectal tumor cell survival [45].

In our work, translational repression was functionally related to downregulation of mitochondrial function, as previously reported in DLBCL by Bhalla et al. in a transcriptomic study [5]. Mitochondria perform central roles in cancer cells, performing several bioenergetic and biosynthetic functions [62], and the mitochondrial proteomic profile is distinct in cancer and non-malignant cells. Requirement of mitochondrial function is necessary for lymphoma progression [63]. Notably, Sharma et al. revealed that mitochondrial respiratory inhibition mimics hypoxic stress and induces RNA editing independently of HIF-1α [24]. The authors hypothesized that hypoxia triggers apolipoprotein B (A3G)-mediated RNA editing by activating a pathway triggered by mitochondrial respiratory inhibition. Indeed, our results showed that hypoxic and physioxic treatment impaired mitochondrial translation in DLBCL cells; in particular, we observed abnormal expression of 10 mitochondrial ribosomal proteins (MRPs) in Ri-1 cells. MRPs are components of the mammalian mitochondrial ribosome, which synthesizes in 13 proteins essential for oxidative phosphorylation [64]. Recently, many researches have demonstrated the abnormal expression of MRPs in various tumors, and importantly, evidence shows an alternative role for MRPs in inducing apoptosis [65,66]. Mitochondrial ribosomal targets were previously found to be differentially regulated in DLBCL under hypoxia [5,67], which is in agreement with our work.

Our results show that prolonged hypoxia significantly inhibited ribosome biogenesis in DLBCL cells. Ribosome, responsible for the translation of information contained in mRNA into the protein, is one of the most conservative structures throughout the evolution. However, its synthesis is one of the most complex biological processes [68]. Ribosomal biogenesis includes the transcription of ribosomal RNA (rRNA), rRNA processing, and production of ribosomal proteins [69]. The aberration in any of these processes may lead to dysregulated ribosome biogenesis, evident in multiple spontaneous cancers [70], including hematological malignancies [69,70,71]. Regarding B-NHL, the role of alterations in ribosome biogenesis is yet to be determined, and only scant evidence suggest a possible correlation with the outcome of diseases [72,73]. Ribosomal proteins are highly responsive to hypoxic stress [74] and several alterations in ribosomal biogenesis were found in our dataset, including downregulation of RPS29, RPS5, RPL6, RPF2, RPL31, BMS1, TSR3, and RPF2. Furthermore, the PPIN analysis of hypoxic Ri-1 cells revealed as many as 51 downregulated proteins involved in ribosome biogenesis. Our proteomic data indicate that DLBCL cell lines converge to a common mechanism with downregulation of proteins involved in ribosome biogenesis, indicating the relevance of these DAPs for a hypoxia-responsive phenotype. Importantly, a similar observation was partially reported by Bhalla et al. in a global transcriptome study [5]. Additionally, the authors observed a differential regulation of ribosomal targets in distinct DLBCL cell lines, which is reflected by our results.

Nearly 20 years ago, Koshiji et al. demonstrated that HIF-1α—rather than other hypoxia-associated genes—that induces the cell cycle arrest [75], which was further confirmed in several genomic and proteomic [31,74,76,77] studies. Cessation of the cell cycle in lymphoma cells was also confirmed in the present work. Moreover, 47 cell proliferation-associated genes were found downregulated in Ri-1 cells under hypoxia, including cyclin CCNB1, and cyclin-dependent kinases CDK14, CDK15, and CDK16, which have significant involvement in the lymphoma pathogenesis. Notably, CDK16 was the most downregulated protein in the entire dataset with a fold change of 195.216 in hypoxic Ri-1 cells. Intriguingly, downregulation of the cell cycle was also noticed in U2904 cells under physioxia; however, in hypoxic Ri-1 cells, the upregulation of M Phase and G2/M transition pathway was observed. The influence of physiological oxygen on the cell cycle is still not well understood; however, studies show mostly acceleration in cell proliferation under physioxia [10,51,78,79]. Notably, in our previously published data, we established that the Ri-1 cell line growth was significantly decreased in physioxia and hypoxia after 96 h of treatment, while the proliferative capability of the remaining cell lines was unchanged [78].

Hypoxia inducible factors (HIFs) perform master roles in the cellular response to hypoxia. HIFs regulate more than 100 genes in response to a decrease in oxygen [80]. Under normoxia, the HIF signaling pathway is inhibited due to HIF-α subunit degradation [81]. The availability of oxygen in normal lymphatic tissues decreased and activation of HIF-1α was observed in normal B-cells during maturation and activation. However, HIF-2α subunit overexpression was observed only following malignant transformation [82]. In our current proteomic data, upregulation of the HIF1α pathway was observed in each cell line under hypoxic stress when compared to normoxia. Moreover, in Ri-1 cells, HIF1α targets, such as those involved in glycolysis, were upregulated, including PDK1 [fold change = 3.3], which is in accordance with other hypoxia-driven studies [83]. Under physioxia, we have not noticed the changes in HIF1 pathways in any cell line; however, some HIF1 related targets were altered, including downregulation of NDUFs in U2932 cells [84]. Interestingly, Miar et al. described for the first a new mechanism of hypoxic immunosuppression via downregulation of the type I IFN pathway [83]. The authors suggested that IFN downregulation is partially dependent on HIF1α. Our data show for the first time the downregulation of the interferon-signaling pathway under both physioxia and hypoxia in DLBCL cells.

It is well established that hypoxia-inducible factors (HIFs) mediate metabolic reprogramming in response to hypoxia [85,86]. Rebuilding of energy metabolism is one of the leading hallmarks of cancer, which enables survival in a hypoxic environment. In our study, cellular amino acid (CAA) metabolism pathway remained the most enriched among upregulated proteins under hypoxia. Therefore, we reported for the first time that lymphoma cells adapt to hypoxic stress through elevating CAA metabolism. This is in line with Zhang et al., who recently suggested the critical role of HIFs in reprograming of cellular amino acid metabolism in glioblastoma [87]. Moreover, our proteomic data showed the main target proteins of HIF-1α were found upregulated, including metabolic targets such as BCAT2, PDK1, HK2, ALDH6A1, and ALDH7A1. Of these, HK2 was described as a key metabolic driver of the DLBCL phenotype [5].

The oxygenation of tissues in vivo is more hypoxic compared to ambient air; however, knowledge of cancer cells responding to a physiological oxygen microenvironment remains a largely overlooked part of cancerogenesis. Here, we made an effort to present the global changes in B-cell proteomes under physioxia. Our data showed that the response of DLBCL cells to physioxia was more cell line specific when compared to hypoxia. The response to the oxidative stress pathway was only one pathway commonly upregulated in each cell line under physioxia.

Interestingly, in our dataset, the enrichment analysis revealed an increase of SUMOylation of transcription factors pathway in Ri-1 cells. The upregulation of small ubiquitin-like modifier (SUMO) is a common posttranslational modification (PTM) in several tumors and is related to tumor development. Accordingly, a high level of SUMOylation was found to be required for cancer cells to survive external stresses, including oxygen deprivation [88]. Interestingly, we identified the NDUF proteins to be important within the PPIN in U2932 cells under physioxia. It was previously hypothesized that suppression of NDUF expression and downregulation of other mitochondrial respiratory chain complex components may be important events contributing to K-Ras-induced mitochondrial dysfunction [89]. Here, we report, for the first time, changes in NDUF synthesis in lymphoma under low oxygen. Concurrently, the variability of enriched pathways made it difficult to draw the precise conclusions regarding the molecular background of the physioxia-induced changes in DLBCL proteome. However, the number of identified DAPs and enriched pathways in each cell line in our dataset suggests that the oxygen level cannot be an omitted parameter when planning in vitro studies with the human cell lines. Moreover, the current studies highlight that cultures in physioxia are more likely to closely mimic microenvironmental effects [32], and following this line of argumentation, more research studying the physioxic impact on overall cell characteristics is urgently needed.

Finally, in search for proteins of the core response to hypoxia and physioxia, a cytoHubba analysis was performed. Our results showed that hypoxic hub proteins were mostly downregulated. The notable hub proteins overlapping in Ri-1 and U2904 cells entirely included proteins involved in ribosome biogenesis (DDX56, MRTO4, RPF2, RSL24D1, and SDAD1). Concurrently, most of the hub proteins identified in Ri-1 and U2904 cells are related to ribosome biogenesis and rRNA processing, suggesting similarity of these two DLBCL cell lines in response to hypoxia. In turn, 13 of the U2932 hub proteins were involved in the cell cycle (mainly mitosis), e.g., CDCA5, CDC23, and CDC45. Importantly, CHEK1 and RRM2 hubs were annotated to the p53 signaling pathway. The selection of hub proteins, which are different for each cell line, additionally confirmed that the response to hypoxia and physioxia is cell line specific; however, the same molecular pathways are differentially regulated in the dataset, Ri-1, U2904, and U2932. DLBCLs are biologically highly heterogeneous tumors; thus, the reason behind the difference in hypoxic and physioxic proteomes of Ri-1, U2932, and U2904 cells, may be due to the metabolic differences between the three cell lines. Similarly, previous gene expression profiling studies revealed differences in several DLBCL cell lines, in their dependencies on metabolic pathways [5,23,90]. Concurrently, the similarity between Ri-1 and U2904 in response to hypoxia was highlighted, which may be associated with the presence of a MYC rearrangement. U2904 and Ri-1 are quickly proliferating lymphomas with MYC gene rearrangement. MYC is a key regulator of cellular metabolism, proliferation, and survival. It is estimated that MYC as a transcription factor can control about 15% of all human genes by inducing or enhancing the expression of previously active genes [91]. Notably, the adaptation to chronic hypoxic stress occurs in part by MYC degradation [92], which leads to changes in the expression of several MYC downstream targets [93].

## 5. Conclusions

To summarize, this report is the first to investigate the influence of decreased oxygen concentration on the global proteomes of DLBCLB-NHL cell lines. Our proteomic data allowed us to define DLBCL phenotypes under hypoxia and physioxia, showing that hypoxia-related response in lymphoproliferative malignancies is complex and highly heterogeneous. However, we were able to indicate several pathways and notable proteins commonly affected by hypoxia in DLBCL. It remains to be investigated whether changes in the proteomes of DLBCL under normoxia and physioxia have functional consequences on lymphoma development and progression, with potential importance for the design of novel treatments. The functional significance of the identified hypoxia-responsive protein targets and pathways must be confirmed in the subsequent biochemical assays. Concurrently we established that the magnitude of hypoxia-driven global changes in the lymphoma proteomes depends on the oxygen concentration used for comparisons; thus, the selection of proper experimental conditions when studying hypoxia should be considered. Finally, given the importance and benefits of maintaining physiological oxygen levels, there is a need to establish more relevant oxygen concentration in vitro studies, especially in potentially life-saving therapies.

## Figures and Tables

**Figure 1 cells-10-02025-f001:**
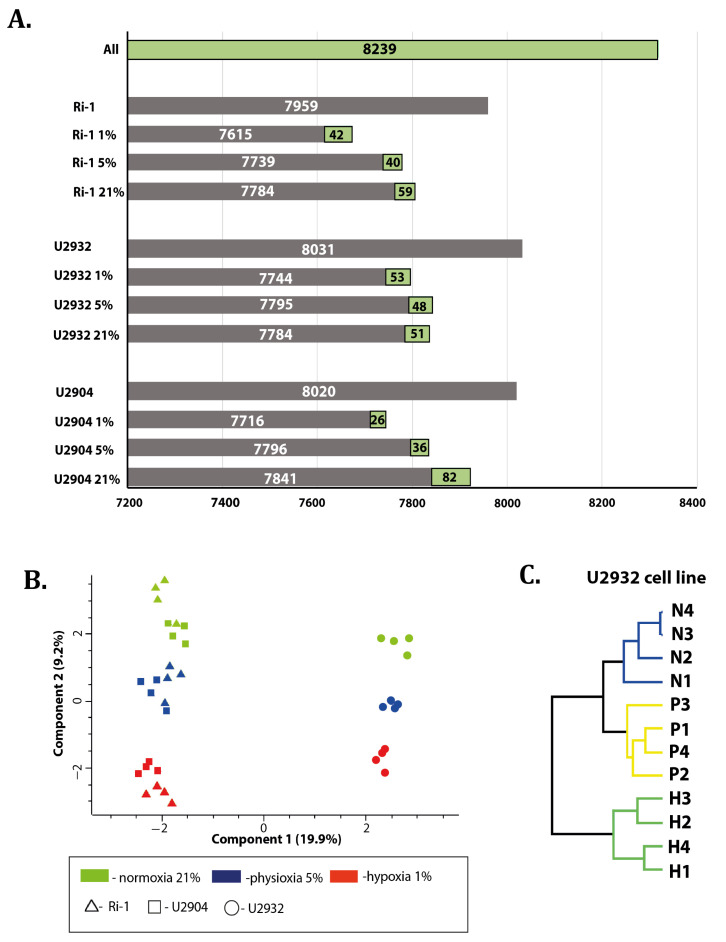
Exploratory analysis. (**A**) The number of proteins identified in the study in each experimental group. The green boxes indicate the number of proteins identified exclusively in the corresponding experimental group. (**B**) Results of principal component analysis (PCA) of log2 intensities values from Ri-1, U2904, and U2932 cell lines (in four biological replicates) under normoxia (21%) physioxia (5%), and hypoxia (1%). (**C**) Dendrogram that displays unsupervised hierarchical clustering analysis between U2932 samples under hypoxia (H, 1% of O_2_), physioxia (P, 5% of O_2_), and normoxia (21% of O_2_). Performed in Perseus software.

**Figure 2 cells-10-02025-f002:**
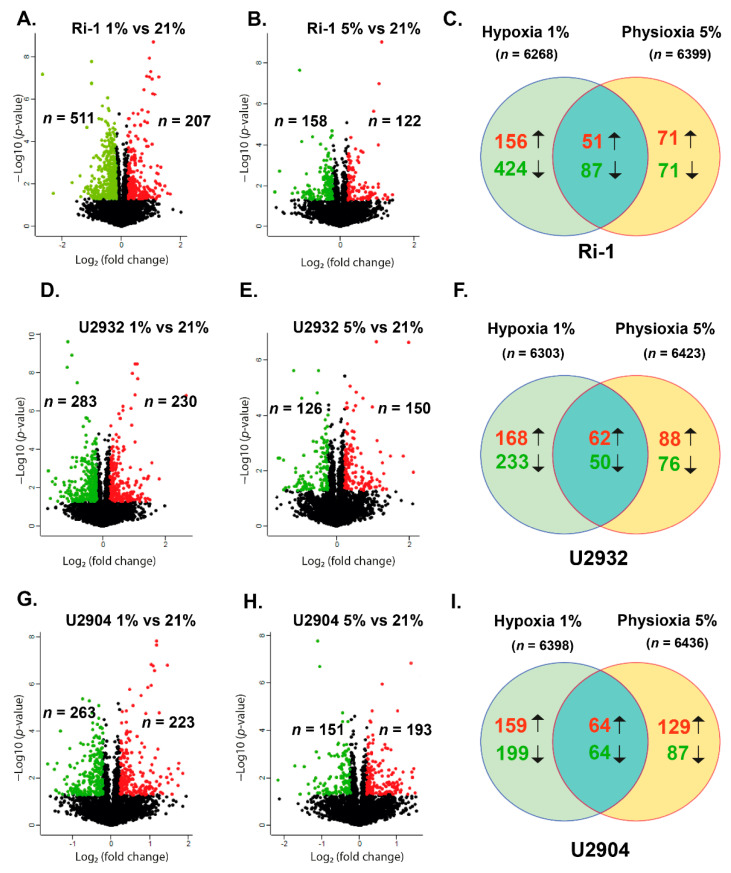
Volcano plots of data from label-free quantification of differentially abundant proteins (DAPs) in (**A**) hypoxic Ri-1 cells, (**B**) physioxic Ri-1 cells, (**D**) hypoxic U2932 cells, (**E**) physioxic U2932 cells, (**G**) hypoxic U2904 cells, and (**H**) physioxic U2904 lymphoma cell lines when compared to normoxia. The number of proteins identified in 100% of biological replicates is provided. Red dots represent upregulated proteins with fold change ≥1.5 while green dots represent downregulated proteins with fold change ≤−1.5, *p*-value cutoff < 0.05. The number of up- and downregulated proteins is presented. Generated in Perseus software. (**C**) Venn diagram showing the overlap between DAPs identified in hypoxia and physioxia in comparison to standard oxygen conditions in Ri-1 cells, (**F**) U2932 cells, and (**I**) U2904 cells. The complete list of DAPs with all details is placed in Appendix A, while lists of shared proteins in hypoxia and physioxia in each cell line are provided in Appendix A.

**Figure 3 cells-10-02025-f003:**
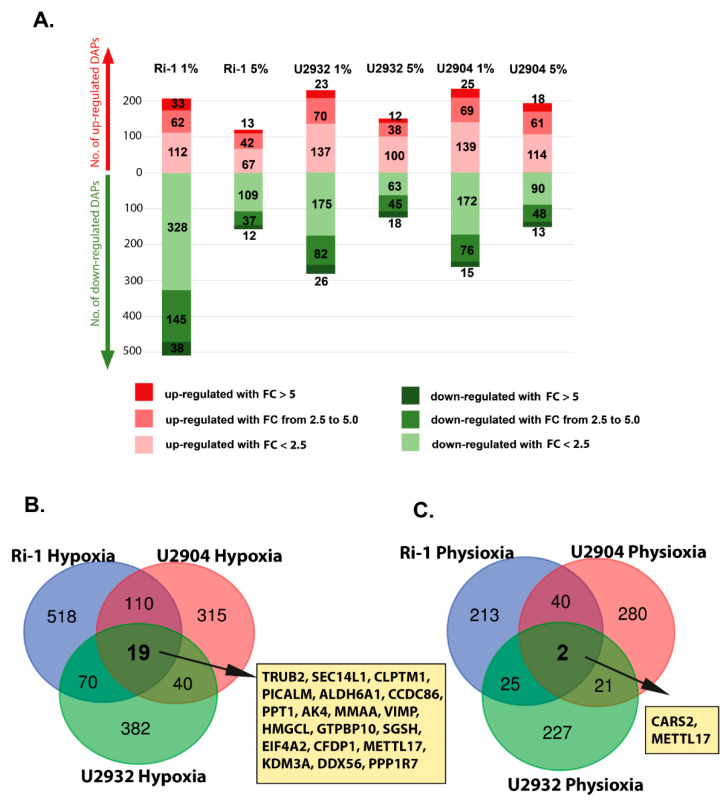
(**A**) Classification of differentially abundant proteins (DAPs) identified in hypoxia (1% of O_2_) and physioxia (5% O_2_) based on fold changes in protein abundance in comparison to control samples maintained in standard oxygen condition, *p* < 0.05. The red and green bars show the number of up- and downregulated proteins, respectively. The color gradation shows increasing fold changes. (**B**) Venn diagrams showing numbers of DAPs overlapping between Ri-1, U2904, and U2932 cell lines in hypoxia (1% of O_2_), and (**C**) physioxia (5% of O_2_). The list of shared proteins is provided in Appendix A.

**Figure 4 cells-10-02025-f004:**
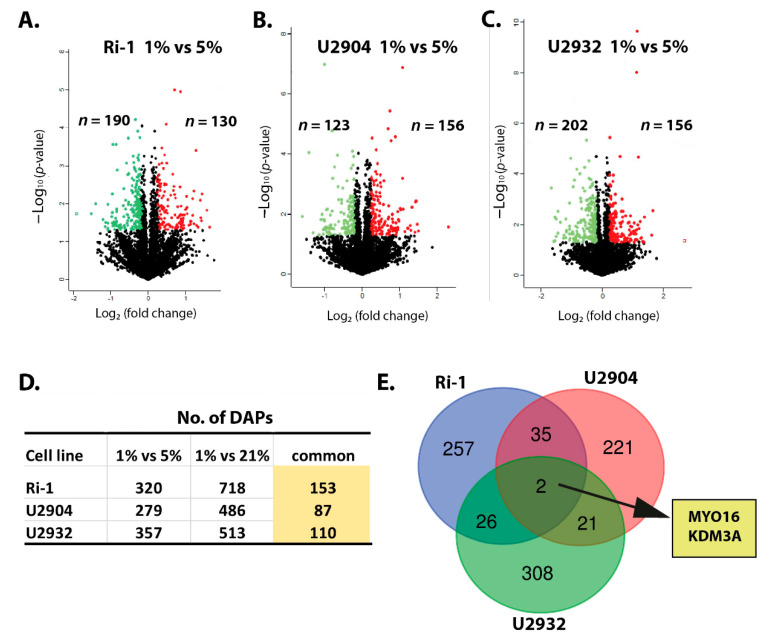
Volcano plots of data from label-free quantification of differentially abundant proteins (DAPs) in hypoxic (1% of O_2_) (**A**) Ri-1 cells, (**B**) U2904 cells, and (**C**) U2932 cells when compared to physioxia (5% of O_2_). Red dots represent upregulated proteins with fold change ≥ 1.5, while green dots represent downregulated proteins with fold change ≤ −1.5, *p*-value cutoff < 0.05. Generated in Perseus software. The complete list of DAPs with all details is placed in Appendix A. (**D**) Venn diagram showing number of DAPs overlapping among Ri-1, U2904, and U2932 cell lines in hypoxia when analyzed versus physioxia. The list of shared proteins is provided in Appendix A. (**E**) The table showing the number of DAPs commonly identified in each cell line when (i) hypoxia compared to physioxia, and (ii) hypoxia compared to normoxia.

**Figure 5 cells-10-02025-f005:**
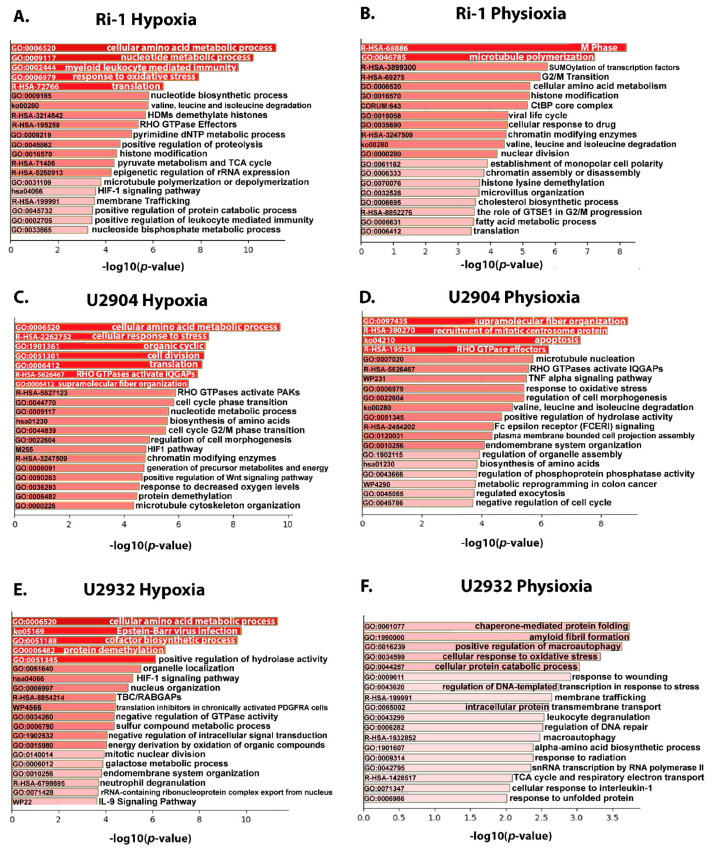
Enrichment analysis of upregulated proteins in hypoxia (1% of O_2_) and physioxia (5% of O_2_) in DLBCL cell lines using the Metascape database. The top 20 clusters with their representative enriched terms (one per cluster) are presented for each experimental condition. (**A**) The enrichment of 207 and 122 significantly upregulated proteins in hypoxia and (**B**) physioxia, respectively in Ri-1 cell line. (**C**) The enrichment of 230 and 150 significantly upregulated proteins in hypoxia and (**D**) physioxia, respectively in U2904 cell line. (**E**) The enrichment of 223 and 193 significantly upregulated proteins in hypoxia and (**F**) physioxia, respectively in U2932 cell line. The x-axis shows the significance, which is the value of −log10(P). Enriched terms are colored by *p*-values.

**Figure 6 cells-10-02025-f006:**
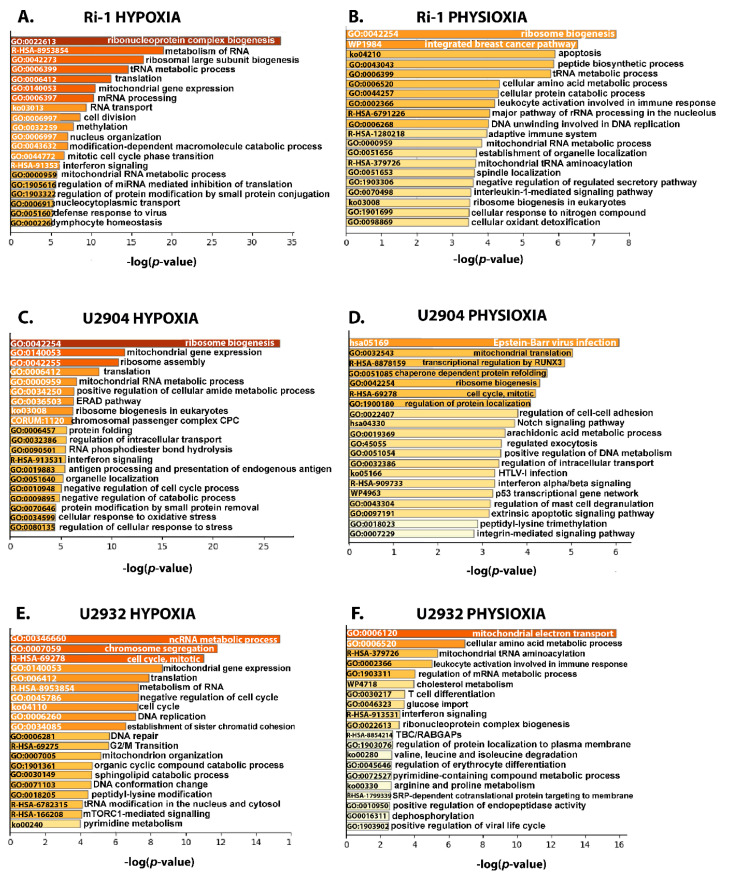
Enrichment analysis of downregulated proteins in hypoxia (1% of O_2_) and physioxia (5% of O_2_) in DLBCL cell lines using the Metascape database. The top 20 clusters with their representative enriched terms (one per cluster) are presented for each experimental condition. (**A**) The enrichment of 511 and 158 significantly downregulated proteins in hypoxia and (**B**) physioxia, respectively, in Ri-1 cell line. (**C**) The enrichment of 283 and 126 significantly downregulated proteins in hypoxia and (**D**) physioxia, respectively, in U2904 cell line. (**E**) The enrichment of 263 and 151 significantly downregulated proteins in hypoxia and (**F**) physioxia, respectively in U2932 cell line. The x-axis shows the significance, which is the value of –log10(P). Enriched terms are colored by *p*-values.

**Figure 7 cells-10-02025-f007:**
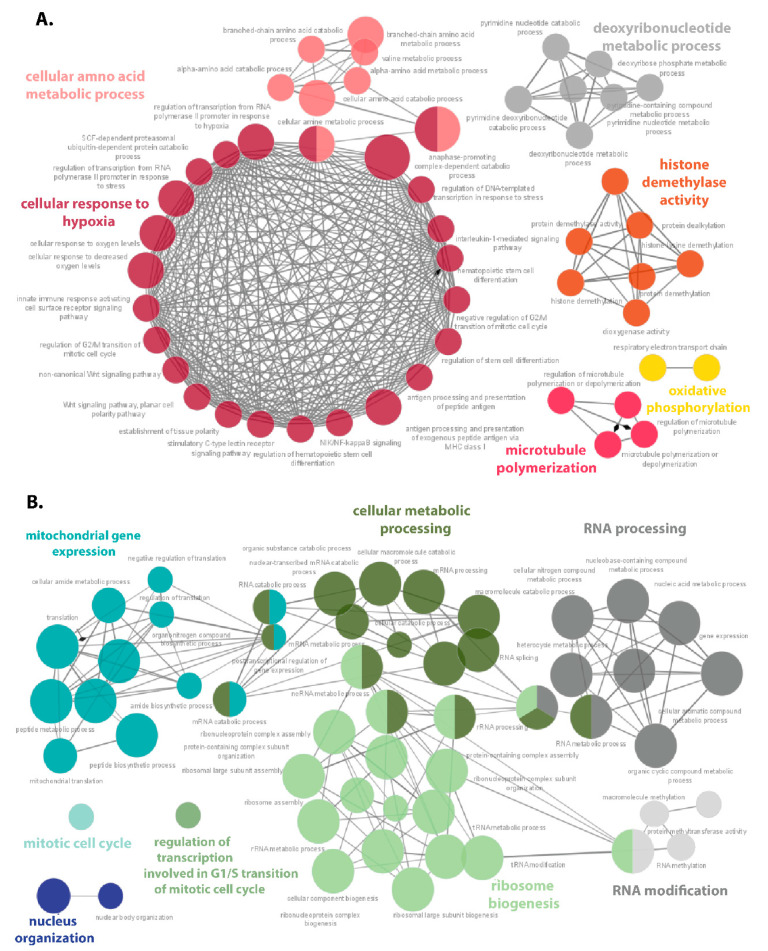
Protein–protein interaction networks of upregulated (**A**), and downregulated (**B**) proteins in Ri-1 cells under hypoxia (1% of O_2_). The analysis was performed and visualized in ClueGO (v.2.5.7) Cytoscape App based on the Gene Ontology–Biological Processes (all updated 18 May 2020). Nodes (circles) indicate the pathway function groups, while edges represent connections between the nodes. Each node color represents a different pathway class that it belongs to. The node size represents the term enrichment significance, while the length of each edge shows the relevancy between two processes, *p* ≤ 0.05. The overlapping areas are the shared pathways of two groups. The details of analysis are provided in Appendix A.

**Figure 8 cells-10-02025-f008:**
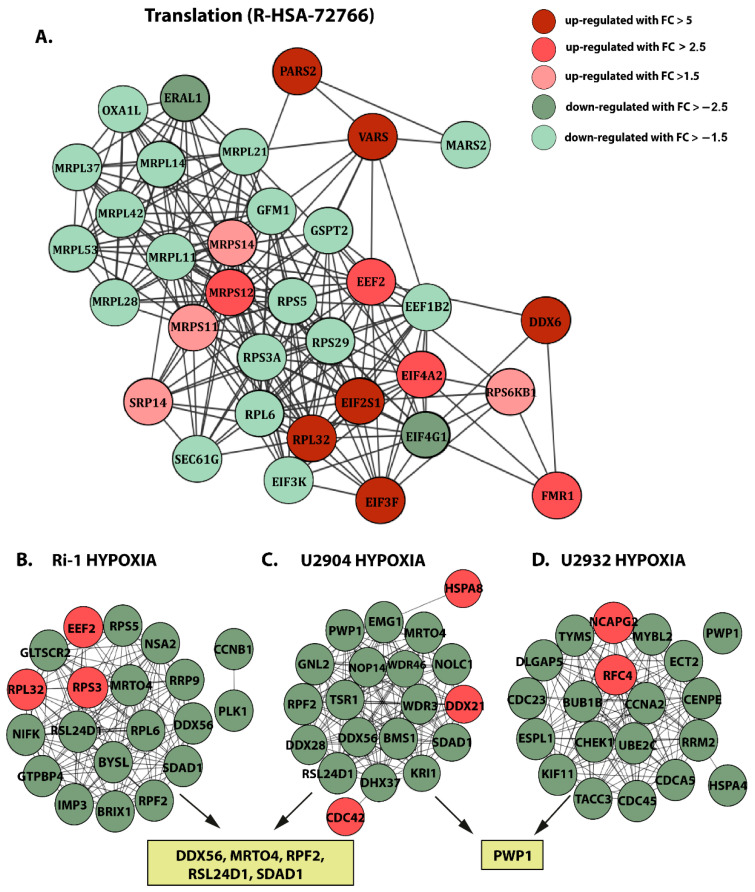
(**A**) Protein–protein interaction network of DAPs identified in Ri-1 cells under hypoxia related to translation (R-HSA-72766). Generated using the stringApp from Cytoscape. (**B**) Top 20 hub proteins identified in hypoxic Ri-1, (**C**) U2904, and (**D**) U2932 cell lines based on the highest degree score. Analyzed and generated using the cytoHubba app from Cytoscape. Red and green nodes represent up- and downregulated proteins, respectively.

**Table 1 cells-10-02025-t001:** The general characteristic of the diffuse large B-cell lymphoma (DLBCL) cell line used in the study. AMP—amplification, R—rearrangements, ABC—activated B-cell like, GCB—germinal center B-cell like, FL—follicular lymphoma, SLL—small lymphocytic lymphoma.

Cell Line	Origin	Subtype	Features
**Ri-1**	transformed SLL (Richter Syndrome)	ABC	MYC ^R^
**U2904**	transformed FL (Kiel centrocytic/centroblastic)	GCB	MYC ^R^, BCL2 ^R^
**U2932**	de novo DLBCL, NOS	ABC	BCL2 ^AMP^

**Table 2 cells-10-02025-t002:** The list of HIF-1α targets being differentially expressed in Ri-1, U2904, and U2932 cell lines under hypoxia. FC- fold change.

Target	Function	Ri-1 (FC)	U2904 (FC)	U2932 (FC)
ALDH6A1	metabolic reprogramming	3.750	2.001	4.112
ALDH7A1		3.689	1.847	
PFKL		3.115	------	4.105
PDK1		3.304	------	2.096
BCAT2		1.529	------	------
PFKFB4		4.396	------	------
HK2		------	2.015	------
BNIP3L		------	6.746	------
LDHA		------	------	−9.452
PFKFB3		------	------	6.548
PGK1		------	------	−5.407
ALDOC		------	------	−3.286
PDCD4	metastasis and invasion	1.879	------	2.370
P4HA1		------	5.630	6.070
CXCR4		------	4.071	------
NPM1		------	3.611	------
LGALS1		------	------	2.468
GPI	angiogenesis	1.590	4.169	
HMOX1			−3.541	−4.303
ZEB1	epithelial- mesenchymal transition	2.210	------	------
VIM	EMT	3.140	------	------
EGLN1	regulation of the hypoxic response	2.253	3.048	------
IRAK4	apoptosis	3.605	------	------

## Data Availability

The mass spectrometry data were deposited at the ProteomeXchange Consortium via the PRIDE partner repository with the dataset identifier: XD026726. Reviewer account details: username: review-er_pxd026726@ebi.ac.uk; password: 6rGpX4.

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
