# Peer review of "Proteomic-Based Analysis of Hypoxia- and Physioxia-Responsive Proteins and Pathways in Diffuse Large B-Cell Lymphoma"

_cells, 2021, doi:10.3390/cells10082025_

Round 1

Reviewer 1 Report

This study titled ‘Proteomic-based analysis of hypoxia- and physioxia-responsive proteins and pathways in diffuse large B-cell lymphoma’ provides a detailed proteomic analysis of three diffuse large B cell Lymphoma (DLBCL) cell lines under different oxygen concentrations for 72 hours. While, this study provides valuable data resource which can be used to learn more about the long-term effects of low oxygen (1%) concentration, there are several recommendations to make this data more relevant to the field (major) and also several minor typographical errors.

Major:

  1. The authors used a hypoxia (1%) treatment of 72 hours. 72 hours is a very long duration of hypoxia. While this may be physiologically relevant, can authors include any references to support that DLBCL cells are exposed to hypoxia for long durations? (Direct evidence by using oxygen sensors) I understand that it is not possible to conduct large scale proteomic analyses for different time points, but it will be important to provide an explanation of why you chose 72 hours time point?
  2. This long-term exposure to hypoxia can be very different to acute hypoxia exposure. While it is important to study the long-term effects, it might be also important to note that in a tissue, at any given time, there would be oxygen gradients and not all cells will be exposed to very low levels of oxygen for long periods. The key question is still whether this time period is pathologically relevant?
  3. Hypoxia can lead to cell cycle arrest in some cases while it can also induce cell proliferation (in some cancers as well as in wound healing). A recent study in mice also showed its highly dynamic and important role of HIF1α during B cell development in bone marrow. So claiming that hypoxia is damaging to normal cells (1st line of abstract) is not accurate. Hypoxia is an evolutionarily conserved signalling mechanism which enables organisms to survive under low oxygen conditions and can also shape differentiation, angiogenesis, and various important processes.
  4. Authors also claim several of their findings could be associated to HIF1α. The regulation of HIF1α is complex and transient. Prolyl hydroxylase domain containing enzymes (PHDs) are transcriptional targets of HIF1α which further hydroxylate and destabilise HIF after several hours of HIF1α stabilisation. This transient behaviour is well-described in literature. HIF2α can become stable after some time but its expression is highly tissue specific. There are several other non-HIF dependent effects of hypoxia too which have been shown in several transcriptomic studies in cancers. Therefore, it is also difficult to attribute the long-term effects solely to HIF1α.
  5. Authors mention that the aggressive effects of DLBCL could be attributed to hypoxia. Again, it is not clear whether the proteomic changes make the cells more aggressive or not. These could be associated with survival which doesn’t necessarily make the cells more aggressive. It might be important to consider what ‘aggressive’ means.
  6. Introduction can benefit from more information about the rationale of choosing physiological oxygen in experimental cultures.
  7. Authors claim in introduction that ‘Hypoxia is a key factor in tumor development’ (Line 55). This statement needs backing with mechanistic studies. Hypoxia is certainly a feature of several solid tumors, but there are several tumors with high vascularisation and oxygenation. Importantly mutations in HIF1α signalling pathway (except for VHL mutations which leads to development of renal carcinoma) are not associated with tumorigenesis.
  8. Authors cite reference 11 claiming hypoxia ‘cancer cells respond to the hypoxic stress by stopping translation [11]’, this statement is again not universally true. There is a specific transcriptional response for hypoxia.
  9. In results section, authors compare hypoxia and physioxia with normoxia (21%). While this is a valid comparison, it might be even more relevant to directly compare hypoxia and physioxia. So the comparisons I would recommend having two comparisons (and separate sections). First: physioxia vs normoxia (to provide evidence that experimental conditions need account for physiological oxygen levels) and a second comparison of physioxia versus hypoxia. This is to compare the physiological level of oxygen versus potential pathologically relevant levels of oxygen. While current comparisons might seem to capture this but the differential analysis will be different.
  10. It might also be worth noting differences between the three cell lines, why they were chosen in the study and their survival fraction under hypoxia. Based on proteomic analysis authors conclude that the Ri-1 cell line is most sensitive to hypoxia. Does this also translate to phenotype? Do these cells phenotypically ‘look’ more stressed/relaxed than other cell lines? Also, authors claim that the differences in response to long-term hypoxia are cell line specific and due to metabolic heterogeneity. It might be important to understand why these cells are responding differently. Do these cell lines have different growth rates? Do they come from different stages of cancer? Do these cell lines harbour known mutations which could potentially explain these variations?
  11. Figure 1B shows a Pearson correlation chart. It might be worth representing the whole data using Principal component analysis (PCA) for all cell lines in one plot. Pearson correlation matrix doesn't provide a confidence interval. That will give us idea about the dominant factors: cell line vs. response to hypoxia/physioxia and it might be interesting to see how the different cell lines differ in their global proteome.
  12. It might be worth commenting on global proteome of the three cell lines.
  13. Authors comment on DAPs which are common to physioxia and hypoxia (Lines 230-232). Are there any proteins which show opposite effect under physioxia and hypoxia?
  14. Authors claim that METT17 is associated with hypoxia but this is erroneous claim. Data shows that METT17 is associated with both physioxia and hypoxia. Hypoxia is a special program and since METT17 is affected in both physioxia and hypoxia, this claim that METT17 is associated with hypoxia is not correct. Consider analysing the data using point 8 (major) and this will highlight hypoxia specific proteins.

Minor:

  1. In abstract, B-NHL is used which is probably referring to DLBCL. Please use DLBCL consistently or define B-NHL
  2. Line 66: Importantly, the authors showed (instead of enclosed)
  3. Line 68-70: Additional recent observations demonstrate further crosstalk of hypoxia and mitochondrial oxidative phosphorylation [15], mTOR [13,16], and BCR [17] pathways in B-NHL. These findings are mainly in B cells and not B-NHL
  4. Line 104: 18h either (instead of neither)
  5. Line 157: U2904 versus normoxicU2904 to identify (instead of identified)
  6. Figure 2 is supposed to be volcano plot. Could you please add what the x-axis represents (log2 fold change or log 10 fold change?) Similarly for y-axis, I am guessing this is -log10 (p value)?
  7. Figure 3a: this figure has green (downregulation) starting from 100 on the y-axis. I think this should start from 0.
  8. Figures 4 and 5 are too wordy and difficult to interpret (similarities and differences). A heatmap showing the pathways and their normalized enrichment score/FDR might be more efficient for direct comparison.
  9. Can you plot proteins belonging to SUMOylation (for example) from the three cell lines and highlight how just one cell line has significant effects but not the others? Similarly for TNF alpha signalling and chaperone mediated protein folding. This would give the reader a visual aspect of what you want to conclude.
  10. Figure 7A has some residual legend on the top of the figure
  11. Line 390: Furthermore, (the is not needed) most of the Ri-1
  12. Line 459: Bhalla et al do not find global transcriptional repression, but instead observed what you have found, repression of some and activation of other programs.
  13. Line 556, knowledge of (instead of how)

Author Response

REVIEWER 1

This study titled ‘Proteomic-based analysis of hypoxia- and physioxia-responsive proteins and pathways in diffuse large B-cell lymphoma’ provides a detailed proteomic analysis of three diffuse large B cell Lymphoma (DLBCL) cell lines under different oxygen concentrations for 72 hours. While, this study provides valuable data resource which can be used to learn more about the long-term effects of low oxygen (1%) concentration, there are several recommendations to make this data more relevant to the field (major) and also several minor typographical errors.

Dear Reviewer,

we would like to thank you for careful reading and widely discussing our article. All comments have been very constructive. We especially appreciate your suggestion to compare hypoxia versus physioxia. We answered all the comments addressed to our work. Please find enclosed a revision, which considers all comments.

Major:

  1. The authors used a hypoxia (1%) treatment of 72 hours. 72 hours is a very long duration of hypoxia. While this may be physiologically relevant, can authors include any references to support that DLBCL cells are exposed to hypoxia for long durations? (Direct evidence by using oxygen sensors) I understand that it is not possible to conduct large scale proteomic analyses for different time points, but it will be important to provide an explanation of why you chose 72 hours time point?

In fact, in the literature, the exposure times used to induce in vitro chronic hypoxia range from few hours to several weeks, with no clear boundaries separating the two.

Thus, before choosing experimental conditions, we have found that culturing cells for 72 h in 1% of O2 concentration is a standard experimental setting when studying long-term hypoxia [1-5].

Unfortunately, we can not cite any in vivo experiments assessing O2 levels within DLCBLs and their environment using, e.g., implanted sensors because of the lack of relevant data.  However, it was established that DLBCL cells meet severe hypoxic conditions during lymphoma progression [6,7]. Rapidly proliferating lymphomas, especially those with MYC rearrangements (including Ri-1 and U2904 cell lines), often lack stromal connective tissues and rich vascularity. They proliferate for a long time under conditions of profound hypoxia reflected in their "starry sky" appearance.

Moreover, DLBCL cells infiltrate frequently to the bone marrow, where hypoxic areas are widely present. Thus DLBCL cells residing in hypoxic niches in the bone marrow can potentially experience a long time. In addition, we were driven by two other studies on hypoxia in DLBCL, where cells were treated in the presence of 1% of oxygen for 48h [8,9].

2. This long-term exposure to hypoxia can be very different to acute hypoxia exposure. While it is important to study the long-term effects, it might be also important to note that in a tissue, at any given time, there would be oxygen gradients and not all cells will be exposed to very low levels of oxygen for long periods. The key question is still whether this time period is pathologically relevant?

Recently, we better understand the need to study the realistic pattern of temporally variable oxygen exposure to the cells and tissues. While ideally controlled laboratory experiments should mimic physiological conditions, the knowledge about the real pattern of oxygen gradient within lymphoma and other hematological malignancies is still limited. Here we presented a first robust analysis to address the differential lymphoma response to physiological and hypoxic oxygen exposure. Even though the experimental conditions are not ideal, this study was planned and performed following applicable standards. However, the Reviewer raised here an essential point of spontaneous fluctuations in long-term hypoxia that cannot be omitted in our work. Thus we supplemented the discussion with issues that had not been mentioned before.

3. Hypoxia can lead to cell cycle arrest in some cases while it can also induce cell proliferation (in some cancers as well as in wound healing). A recent study in mice also showed its highly dynamic and important role of HIF1α during B cell development in bone marrow. So claiming that hypoxia is damaging to normal cells (1st line of abstract) is not accurate. Hypoxia is an evolutionarily conserved signalling mechanism which enables organisms to survive under low oxygen conditions and can also shape differentiation, angiogenesis, and various important processes

We rephrased the abovementioned  sentence in accordance with the Reviewer’s suggestion.

4. Authors also claim several of their findings could be associated to HIF1α. The regulation of HIF1α is complex and transient. Prolyl hydroxylase domain containing enzymes (PHDs) are transcriptional targets of HIF1α which further hydroxylate and destabilise HIF after several hours of HIF1α stabilisation. This transient behaviour is well-described in literature. HIF2α can become stable after some time but its expression is highly tissue specific. There are several other non-HIF dependent effects of hypoxia too which have been shown in several transcriptomic studies in cancers. Therefore, it is also difficult to attribute the long-term effects solely to HIF1α.

We agree that other HIFs independent mechanisms may regulate hypoxic response. However, hypoxic response mechanisms in lymphomas are not largely documented; the large-scale transcriptomic and proteomic analyses are lacking. Thus in the discussion, we tried to associates our main findings with the current state of knowledge.

5. Authors mention that the aggressive effects of DLBCL could be attributed to hypoxia. Again, it is not clear whether the proteomic changes make the cells more aggressive or not. These could be associated with survival which doesn’t necessarily make the cells more aggressive. It might be important to consider what ‘aggressive’ means.

We changed the above statement. Instead of that, we conclude: It remains to be investigated whether changes in the proteomes of DLBCL under normoxia and physioxia have functional consequences on their progression.

 6. Introduction can benefit from more information about the rationale of choosing physiological oxygen in experimental cultures.

The Introduction have been completed about the arguments for performing in vitro study under physiological oxygen concentration. Please see lines 45-57.

7. Authors claim in introduction that ‘Hypoxia is a key factor in tumor development’ (Line 55). This statement needs backing with mechanistic studies. Hypoxia is certainly a feature of several solid tumors, but there are several tumors with high vascularisation and oxygenation. Importantly mutations in HIF1α signalling pathway (except for VHL mutations which leads to development of renal carcinoma) are not associated with tumorigenesis.

We used generalization in that statement, which is not entirely true. We rephrased the abovementioned sentence. Please, see lines: 73-74.

8. Authors cite reference 11 claiming hypoxia ‘cancer cells respond to the hypoxic stress by stopping translation [11]’, this statement is again not universally true. There is a specific transcriptional response for hypoxia.

We have change above statement, as well as corresponding reference. Please see lines 77-79 and new reference no. 21 .

9. In results section, authors compare hypoxia and physioxia with normoxia (21%). While this is a valid comparison, it might be even more relevant to directly compare hypoxia and physioxia. So the comparisons I would recommend having two comparisons (and separate sections). First: physioxia vs normoxia (to provide evidence that experimental conditions need account for physiological oxygen levels) and a second comparison of physioxia versus hypoxia. This is to compare the physiological level of oxygen versus potential pathologically relevant levels of oxygen. While current comparisons might seem to capture this but the differential analysis will be different.

Usually, in hypoxia and physioxia driven studies, cells are cultured at specific oxygen concentrations relative to normoxia (21%). However, considering that hypoxia is much closer to the physioxic range than normoxia, we found the reviewer comment very valuable and performed additional analysis suggested. Next, we compared and commented on the results obtained in these two comparisons. Next, we prepared new figures: Figure 2 and Figure S1 and described results in a new section entitled: Hypoxic Treatment Exhibit Lower Magnitude of Changes When Compared With Physioxia. Finally, we added the eight new lists with up-and down-regulated DAPs in each cell line to Table S2, and we added a new list with DAPs overlapping between cell lines (hypoxia versus normoxia) to Table S3.

10. It might also be worth noting differences between the three cell lines, why they were chosen in the study and their survival fraction under hypoxia. Based on proteomic analysis authors conclude that the Ri-1 cell line is most sensitive to hypoxia. Does this also translate to phenotype? Do these cells phenotypically ‘look’ more stressed/relaxed than other cell lines? Also, authors claim that the differences in response to long-term hypoxia are cell line specific and due to metabolic heterogeneity. It might be important to understand why these cells are responding differently. Do these cell lines have different growth rates? Do they come from different stages of cancer? Do these cell lines harbour known mutations which could potentially explain these variations?

The changes in the proteomes under hypoxia may differ because cell lines represent three biologically distinct neoplasms within one DLBCL group, please see Table.

Table 1. The general characteristic of the DLBCL cell line used in the study.

Cell line

Origin

Type

Features

Ri-1

transformed SLL (Richter Syndrome)

ABC

MYCR

U2904

transformed FL (Kiel centrocytic/centroblastic)

GCB

MYCR, BCL2R

U2932

de novo DLBCL,NOS

ABC

BCL2 AMP

AMP- amplification, R-rearragments.

Observed differences in the response of cell lines to hypoxia may partially be dependent on MYC oncogene. The PCA analysis of the entire dataset revealed the similarity in the proteomes of Ri-1 and U2904 cell lines.

U2904 and Ri-1 are quickly proliferating lymphomas with MYC gene rearrangement. MYC is a key regulator of cellular metabolism, proliferation, and survival. It is estimated that MYC as a transcription factor can control about 15% of all human genes by inducing or enhancing the expression of previously active genes. Notably, the adaptation to chronic hypoxic stress occurs in part by MYC degradation [1]. It was observed that knockdown of HIF-2α increased levels of c-Myc and its several downstream targets in chronic hypoxia, indicating that HIF-2α may function to downregulate c-Myc.

Moreover, in our previously published data, we established that Ri-1 cell line growth was significantly decreased in physioxia (5% od O2) and hypoxia (1%) after 96h of treatment, while the proliferative capability of the remaining cell lines was unchanged. Please see lines 702-710.

 11. Figure 1B shows a Pearson correlation chart. It might be worth representing the whole data using Principal component analysis (PCA) for all cell lines in one plot. Pearson correlation matrix doesn't provide a confidence interval. That will give us idea about the dominant factors: cell line vs. response to hypoxia/physioxia and it might be interesting to see how the different cell lines differ in their global proteome.

The PCA analysis of entire dataset was included into supplementary material. Please see Figure S1 as well as the lines  218-221 within the main text.

12. It might be worth commenting on global proteome of the three cell lines.

We tried to make our work concise and focused on emphasizing the most original findings of the influence of hypoxia and physioxia on DLBCL proteomes. At the same time, we provided to readers the complete list of identified proteins and row files uploaded to the database to perform further independent analysis. In addition, there is a work of our institutional colleagues where the proteomes of DLBCL were greatly described (please see: Deeb SJ et al. Super-SILAC allows classification of diffuse large B-cell lymphoma subtypes by their protein expression profiles. Mol Cell Proteomics. 2012 May;11(5):77-89.)

13. Authors comment on DAPs which are common to physioxia and hypoxia (Lines 230-232). Are there any proteins which show opposite effect under physioxia and hypoxia?

   Vast majority of DAPs showed the same direction of changes, please see representative examples below.

14. Authors claim that METT17 is associated with hypoxia but this is erroneous claim. Data shows that METT17 is associated with both physioxia and hypoxia. Hypoxia is a special program and since METT17 is affected in both physioxia and hypoxia, this claim that METT17 is associated with hypoxia is not correct. Consider analysing the data using point 8 (major) and this will highlight hypoxia specific proteins.

The protein dysregulated in both: physioxia and hypoxia can not be regulated by hypoxia. We intended to highlight DAPs with changed abundances under both treatments. We also decided to remove the sentence: These proteins may be involved in hypoxia-induced malignant DLBCLB-NHL phenotype.

Minor:

  1. In abstract, B-NHL is used which is probably referring to DLBCL. Please use DLBCL consistently or define B-NHL

We have changed the term B-NHL to DLBCL where appropriate.

2. Line 66: Importantly, the authors showed (instead of enclosed)

The sentence have been reworded.

3. Line 68-70: Additional recent observations demonstrate further crosstalk of hypoxia and mitochondrial oxidative phosphorylation [15], mTOR [13,16], and BCR [17] pathways in B-NHL. These findings are mainly in B cells and not B-NHL

 The sentence have been reworded.

 4. Line 104: 18h either (instead of neither)

   The sentence have been corrected.

 5. Line 157: U2904 versus normoxicU2904 to identify (instead of identified)

Above mistake was corrected.

6. Figure 2 is supposed to be volcano plot. Could you please add what the x-axis represents (log2 fold change or log 10 fold change?) Similarly for y-axis, I am guessing this is -log10 (p value)?

               X-axis represents log2 fold change, while x-axis is -log10 p-value. The axes in the figures have been re-labeled.

7. Figure 3a: this figure has green (downregulation) starting from 100 on the y-axis. I think this should start from 0.

Thank you for your accuracy- the mistake have been corrected.

 8. Figures 4 and 5 are too wordy and difficult to interpret (similarities and differences). A heatmap showing the pathways and their normalized enrichment score/FDR might be more efficient for direct comparison.

 A similar way of illustrating the enrichment analysis results performed in Metascape has already been used in cancer investigation studies (please see references below). We prefer to leave Figures in a present form.

Zhou Q, Zhang F, He Z, Zuo MZ. E2F2/5/8 Serve as Potential Prognostic Biomarkers and Targets for Human Ovarian Cancer. Front Oncol. 2019 Mar 22;9:161.

Lyu L, Xiang W, Zhu JY, Huang T, Yuan JD, Zhang CH. Integrative analysis of the lncRNA-associated ceRNA network reveals lncRNAs as potential prognostic biomarkers in human muscle-invasive bladder cancer. Cancer Manag Res. 2019 Jul 4;11:6061-6077.

Ju Q, Zhao YJ, Dong Y, Cheng C, Zhang S, Yang Y, Li P, Ge D, Sun B. Identification of a miRNA-mRNA network associated with lymph node metastasis in colorectal cancer. Oncol Lett. 2019 Aug;18(2):1179-1188

Can you plot proteins belonging to SUMOylation (for example) from the three cell lines and highlight how just one cell line has significant effects but not the others? Similarly for TNF alpha signalling and chaperone mediated protein folding. This would give the reader a visual aspect of what you want to conclude.

Figure 8 and Figure S3 shows the hub proteins, which in a protein-protein interaction network system have remarkably higher interaction relations. They were selected among the all DAPs identified for a particular cell line, separately for hypoxia (Figure 8) and physioxia (Figure S3); thus, they differ between cell lines. Interestingly, Ri-1 and U2904 cells share some of the DAPs under hypoxia.  Our work already has eight main Figures plus three supplementary Figures; thus, we would not like to post subsequent figures to avoid the overloading of the manuscript.

  1. Figure 7A has some residual legend on the top of the Figure.

The unnecessary item has been deleted.

  1. Line 390: Furthermore, (the is not needed) most of the Ri-1

The sentence have been corrected.

  1. Line 459: Bhalla et al do not find global transcriptional repression, but instead observed what you have found, repression of some and activation of other programs. This regulation in protein translation machinery is coupled with reduction in mitochondrial function and resistance of cell growth under hypoxic stress.

Bhalla et al. in their paper has written: Our data delineate that hypoxic stress results in global repression of protein translation with selective stimulation of clinically correlated hypoxia targets such as GLUT1, HK2, and CTY-C. Nevertheless, their data showing the switching of translation. We have to change the sentence as followed: Bhalla et al. observed an oxygen-regulated switch in the protein synthesis machinery that activation of HIF1α resulted in global translation repression during hypoxic stress in DLBCL [5], which was partially confirmed in our work

 Line 556, knowledge of (instead of how)

The sentence have been corrected.

Reviewer 2 Report

In this manuscript Duś-Szachniewicz et al. report a proteomics study in three diffuse large B-cell lymphoma (DLBCL) cell lines kept under 21%, 5% and 1% Oxygen and identified differentially expressed proteins between each condition and cell line. Furthermore, they analyze the different pathways that these differentially abundant proteins (DAPs) belong in respect to different oxygen levels. This is a well written paper, that presents interesting data on the proteome of DLBCL cell lines exposed at different oxygen conditions. However, there are issues to be mitigated before it can be accepted.

1) In their comparisons the authors indeed compare and show the common group of proteins (DAPs) between 5% Oxygen and 1% Oxygen when compared to the normoxic group. However, it is also very interesting and valuable for the authors to show (and analyse) in a different table the DAPs that differ between 5% Oxygen and 1% Oxygen. This is important, since it is known that in different low oxygen conditions the oxygen-sensitive enzymes that control HIFs (namely PHDs and FIH) posses different Km for oxygen and differentially regulate HIFs depending on oxygen concentration.

2) The main novelty of this article is the proteomic study of DLBCL cell lines at different oxygen concentrations. However, there are quantitative studies from DLBCL patient samples (L.-M. Fornecker et al, Sci.Rep. 2019 ; L. E. van der Meeren et al, PLOS One 2019). It will greatly improve the paper the comparison and discussion of the authors’ results with those from patient datasets. This comparison will also give information on which of the conditions approaches reality.

3) Since the authors investigate DLBCL cell lines under different oxygen concentrations and detect HIF targets in their data (as mentioned in lines 552-553) it will be helpful to provide also a table of DAPs that are confirmed HIF targets.

Minor issues:

1) in supplementary data there are two files named Table S4 one should be S3.

2) In reference list beginning from line 690: there is duplicate and different numbering of references. It should be mitigated.

Author Response

In this manuscript Duś-Szachniewicz et al. report a proteomics study in three diffuse large B-cell lymphoma (DLBCL) cell lines kept under 21%, 5% and 1% Oxygen and identified differentially expressed proteins between each condition and cell line. Furthermore, they analyze the different pathways that these differentially abundant proteins (DAPs) belong in respect to different oxygen levels. This is a well written paper, that presents interesting data on the proteome of DLBCL cell lines exposed at different oxygen conditions. However, there are issues to be mitigated before it can be accepted.

Dear Reviewer, We would like to thank you for taking the time to carefully reading and discussing our article. In particular, we appreciate the positive feedback from the Reviewer. We answered all the comments addressed to our work. Please find enclosed a revision, which considers the reviewers’ comments.

1. In their comparisons the authors indeed compare and show the common group of proteins (DAPs) between 5% Oxygen and 1% Oxygen when compared to the normoxic group. However, it is also very interesting and valuable for the authors to show (and analyse) in a different table the DAPs that differ between 5% Oxygen and 1% Oxygen. This is important, since it is known that in different low oxygen conditions the oxygen-sensitive enzymes that control HIFs (namely PHDs and FIH) posses different Km for oxygen and differentially regulate HIFs depending on oxygen concentration.

We are glad for that suggestion, thanks to which we could increase the value of the work. We performed the new comparisons between physiological (5%) and hypoxic (1%) oxygen conditions for each cell line. Next, we prepared new figures: Figure 2 and Figure S1 and described results in a new section entitled: Hypoxic Treatment Exhibit Lower Magnitude of Changes When Compared With Physioxia. Finally, we added the eight new lists with up-and down-regulated DAPs in each cell line to Table S2, and we added a new list with DAPs overlapping between cell lines (hypoxia versus normoxia) Table S3.

2) The main novelty of this article is the proteomic study of DLBCL cell lines at different oxygen concentrations. However, there are quantitative studies from DLBCL patient samples (L.-M. Fornecker et al, Sci.Rep. 2019 ; L. E. van der Meeren et al, PLOS One 2019). It will greatly improve the paper the comparison and discussion of the authors’ results with those from patient datasets. This comparison will also give information on which of the conditions approaches reality.

This comment is of particular importance to us. According to the Reviewer suggestion, we compared the previously published data of Fornecker et al. Multi-omics dataset to decipher the complexity of drug resistance in diffuse large B-cell lymphoma Scientific Reports, 2019, 9, 895. with our list of proteins significantly changed under hypoxia (vs. normoxia). Surprisingly, we were able to find several targets that were significantly changed in both: chemoresistant DLBCL patients and DLBCL cell lines under hypoxia.

Shortly, AK4 and GTPBP10 were commonly found in all DLBCL lines and chemoresistant clinical samples.  FAM98A, DHX30, PML, and CTSZ were found in patients and Ri-1 and U2904 cell lines.  GNL3, CLUH, PARP4, HSDL1, and NOB1 were identified in patients and Ri-1 and U2932 cells, while TAP1, RPS19, IFI35, and MYO1G were found to be changed in patients, U2904, and U2932 cells. Next,  25, 21, and 23 of DAPs were shared between patients and Ri-1, U2904, and U2932 cells, respectively. This data may indicate potentially significant crosstalk between hypoxia and lymphoma progression; thus, we believe that this issue will be worth analyzing in a new article, especially since we have unpublished proteomic data regarding DLBCL level of 8.000 proteins with accompanying follow-up data. Thank you in advance for understanding.

3) Since the authors investigate DLBCL cell lines under different oxygen concentrations and detect HIF targets in their data (as mentioned in lines 552-553) it will be helpful to provide also a table of DAPs that are confirmed HIF targets.

We have prepared the list of DAPs identified in our dataset, which are confirmed HIF targets by Dengler V et al. “Transcriptional regulation by hypoxia-inducible factors.” Critical reviews in biochemistry and molecular biology, 2014, 49,1, HIF-1-alpha transcription factor network (M255), and HIF-1 signaling pathway (hsa04066) provided by Metascape. Please see the new Table 1.      

Minor issues:

1) in supplementary data there are two files named Table S4 one should be S3.

The mistake have been corrected.

2) In reference list beginning from line 690: there is duplicate and different numbering of references. It should be mitigated.

The new reference list with the proper numbering has been prepared.

Round 2

Reviewer 1 Report

Reviewer thanks the authors for carefully updating the manuscript and answer the comments. I have a few minor clarifications and typos which are listed below:

Key clarification:

1) In methods section: Line 186: 'The ratio was calculated by dividing the mean TPA values of the lymphoma group with the mean TPA values of the control  group.' Can you clarify what the lymphoma group and what the control group imply in this context?

2) I think figure 1b and 1c show similar information and it might be better to replace figure 1b with PCA plot as that is providing a richer information. P.S., I can't see the supplementary images. 

3) It might worth discussing some proteins which are highly affected such as LRRC15 and CRYZ

Minor typos: 

1) Line 59: I think the authors meant to write in vitro studies in 'raise essential questions on the relevance of the in vivo studies'

2) Line 57: ' Surprisingly, the data ... in physiological oxygen IS lacking'

3) Line 226, there is repetition of 'For protein identification data see Table S1'

4) Line 334: Response to hypoxia when compared with physioxia

5) Colours in Figure 5 and 6 (enrichment plots) do not add any new information to the data as the x-axis already shows the -log10 p value. From a readers' point of view, it might better to exclude the colours or make sure that the text in dark red background is changed to white instead of black. 

6) Line 719: and progression, (no .)

7) Line 724: when studying hypoxia should be considered. 

8) line 726: in in vitro studies (in vitro italics)

Author Response

Dear Reviewer,

thank you again for taking the time to carefully reading and improving our article. We appreciate the positive feedback regarding our revised work. At the same time we are glad for that suggestions, thanks to which we could increase the value of the work. Below we answered the minor comments addressed to our manuscript.

Reviewer thanks the authors for carefully updating the manuscript and answer the comments. I have a few minor clarifications and typos which are listed below:

Key clarification:

  • In methods section: Line 186: 'The ratio was calculated by dividing the mean TPA values of the lymphoma group with the mean TPA values of the control  group.' Can you clarify what the lymphoma group and what the control group imply in this context?

We clarified how to calculate the ratio within the study groups; please see lines 183-186.

  • I think figure 1b and 1c show similar information and it might be better to replace figure 1b with PCA plot as that is providing a richer information. P.S., I can't see the supplementary images. 

We replaced Figure 1B with Figure S1 (PCA plot) in accordance with the Reviewers’ suggestion. At the same time Figure 1B was transferred to Supplementary Material.

  • It might worth discussing some proteins which are highly affected such as LRRC15 and CRYZ

We have prepared some essential information about the LRRC15 and CRYZ proteins; however we are not sure whether to include this lines in the manuscript. The role of above proteins in cancer is elusive, as well as their behavior under hypoxia. Moreover, these proteins do not belong to any of the pathways discussed in this work. Therefore we have the impression that the newly prepared part of the text does not fit in with the rest of the discussion. We count on your understanding in this matter.

Newly prepared fragment:

Zeta-cristallin (CRYZ) was the most highly up-regulated protein under hypoxia, with a fold change of 9.545. In human, it can be found at enzymatic levels practically ubiquitously in the various organs and tissues. Moreover, like many other crystallins, CryZ is a moonlighting protein, performing more than one physiologically relevant biochemical or biophysical function within one polypeptide chain [1]. The implications in cancer of the many functions of CryZ are discussed, however, CryZ was identified as a Bcl-2- and Bcl-xL-binding protein involved in leukemia cell resistance to apoptosis [2]. In turn, LRRC15 was found to be the most significantly down-regulated protein under hypoxia (fold change= 48.331). LRRC15 is a member of the leucine-rich repeat (LRR) superfamily, which was recently found as a new marker of cancer-associated fibroblasts and cancers of mesenchymal origin [3]. It was suggested to be involved in protein‐protein and protein‐extracellular matrix interactions, as well as signal transduction for various cellular processes [4]. However, the role of LRRC15 in lymphoma has not been delineated. Importantly, LRRC15 was proposed as a stromal target for antibody‐drug conjugates in sarcomas, breast cancer, and glioblastoma, significantly reducing tumor volume [5]. The link with hypoxia has not been discovered so far.

  1. Lulli, M.; Nencioni, D.; Papucci, L.; Schiavone, N. Zeta-crystallin: a moonlighting player in cancer. Cell Mol. Life Sci. 2020, 77, 965-976.
  2. Lapucci, A.; Lulli, M.; Amedei, A.; Papucci, L.; Witort, E.; Di Gesualdo, F.; Bertolini, F.; Brewer, G.; Nicolin, A.; Bevilacqua, A. zeta-Crystallin is a bcl-2 mRNA binding protein involved in bcl-2 overexpression in T-cell acute lymphocytic leukemia. FASEB J. 2010, 24, 1852-1865
  3. Purcell, J.W.; Tanlimco, S.G.; Hickson, J.; Fox, M.; Sho, M.; Durkin, L.; Uziel, T.; Powers, R.; Foster, K.; McGonigal, T.; et.al . LRRC15 Is a Novel Mesenchymal Protein and Stromal Target for Antibody-Drug Conjugates. Cancer Res. 2018, 78, 4059-4072.
  4. Gieniec, K.A.; Butler, L.M.; Worthley, D.L.; Woods, S.L. Cancer-associated fibroblasts-heroes or villains? Br. J. Cancer. 2019, 121, 293-302
  5. Demetri, G.D.; Luke, J.J.; Hollebecque, A.; Powderly, J.D. 2nd.; Spira, A.I.; Subbiah, V.; Naumovski, L.; Chen, C.; Fang, H.; Lai, D.W. et al. First-in-Human Phase I Study of ABBV-085, an Antibody-Drug Conjugate Targeting LRRC15, in Sarcomas and Other Advanced Solid Tumors. Clin. Cancer Res. 2021, 27, 3556-3566.

Minor typos: 

  • Line 59: I think the authors meant to write in vitro studies in 'raise essential questions on the relevance of the in vivo studies'

The above mistake has been corrected.

  • Line 57: ' Surprisingly, the data ... in physiological oxygen IS lacking'

The sentence has been completed.

  • Line 226, there is repetition of 'For protein identification data see Table S1'

The repetition has been deleted.

  • Line 334: Response to hypoxia when compared with physioxia

The above mistake has been corrected.

  • Colours in Figure 5 and 6 (enrichment plots) do not add any new information to the data as the x-axis already shows the -log10 p value. From a readers' point of view, it might better to exclude the colours or make sure that the text in dark red background is changed to white instead of black. 

We changed the color of the letters to white, and we enlarged fonts in Figures 5 and 6 to improve the figures' readability.

  • Line 719: and progression, (no .)

The redundant dot has been removed.

  • Line 724: when studying hypoxia should be considered.

Above sentence has been reworded.

  • line 726: in in vitro studies (in vitro italics)

The term in vitro was written in italics.

Reviewer 2 Report

The authors tried and responded to most of my concerns.

Author Response

Dear Reviewer,

thank you again for taking the time to carefully reading our article. We appreciate the positive feedback regarding our revised work. At the same time we are glad for that suggestions, thanks to which we could increase the value of this work.

With Best Regards!

Kamila Dus-Szachniewicz with co-authors.